# Bio-Applications of Multifunctional Melanin Nanoparticles: From Nanomedicine to Nanocosmetics

**DOI:** 10.3390/nano10112276

**Published:** 2020-11-17

**Authors:** Alexandra Mavridi-Printezi, Moreno Guernelli, Arianna Menichetti, Marco Montalti

**Affiliations:** 1Department of Chemistry “Giacomo Ciamician”, University of Bologna, Via Selmi 2, 40126 Bologna, Italy; alexandra.mavridi2@unibo.it (A.M.-P.); moreno.guernelli2@unibo.it (M.G.); arianna.menichetti2@unibo.it (A.M.); 2Tecnopolo di Rimini, Via Campana 71, 47922 Rimini, Italy

**Keywords:** nanomedicine, polydopamine, melanin, biocompatible, theranostics, photothermal therapy, photodynamic therapy, drug delivery, bioimaging, nanotoxicity

## Abstract

Bioinspired nanomaterials are ideal components for nanomedicine, by virtue of their expected biocompatibility or even complete lack of toxicity. Natural and artificial melanin-based nanoparticles (MNP), including polydopamine nanoparticles (PDA NP), excel for their extraordinary combination of additional optical, electronic, chemical, photophysical, and photochemical properties. Thanks to these features, melanin plays an important multifunctional role in the design of new platforms for nanomedicine where this material works not only as a mechanical support or scaffold, but as an active component for imaging, even multimodal, and simple or synergistic therapy. The number of examples of bio-applications of MNP increased dramatically in the last decade. Here, we review the most recent ones, focusing on the multiplicity of functions that melanin performs in theranostics platforms with increasing complexity. For the sake of clarity, we start analyzing briefly the main properties of melanin and its derivative as well as main natural sources and synthetic methods, moving to imaging application from mono-modal (fluorescence, photoacoustic, and magnetic resonance) to multi-modal, and then to mono-therapy (drug delivery, anti-oxidant, photothermal, and photodynamic), and finally to theranostics and synergistic therapies, including gene- and immuno- in combination to photothermal and photodynamic. Nanomedicine aims not only at the treatment of diseases, but also to their prevention, and melanin in nature performs a protective action, in the form of nanopigment, against UV-Vis radiations and oxidants. With these functions being at the border between nanomedicine and cosmetics nanotechnology, recently examples of applications of artificial MNP in cosmetics are increasing, paving the road to the birth of the new science of nanocosmetics. In the last part of this review, we summarize and discuss these important recent results that establish evidence of the interconnection between nanomedicine and cosmetics nanotechnology.

## 1. Introduction

The advent and development of nanoscience and nanotechnology gave rise to a meteoric evolution in sciences related to human health preservation, including medicine: structurally and functionally tailored nanostructures emerged as promising platforms for the development of new diagnostic and therapeutic tools, especially, but not exclusively, for cancer treatment up to the clinical level [1,2,3,4]. Nanoparticles (NP) offer, as advantages, the simplicity of the control and tuning of size and morphology as well as the versatility of surface modification and compartmentalized loading, to enhance selective recognition and specific action on selected targets [5,6,7].

Multi-functionality is a crucial breakthrough feature of these nanoagents that can be simply engineered to achieve combined simultaneous diagnostic and therapeutic activity in a theranostic approach that, in the most advanced developments, joins multi-modal imaging (e.g., fluorescence, FI; magnetic resonance, MRI; photoacoustic; PAI) to multi-therapy (e.g., chemo-, CT; photothermal, PTT; photodynamic, PDT; gene- or immuno-) in a combined or even synergistic mode [8].

Starting from the most elementary function, NP are often simply, but very importantly, exploited as nanocarrier able to transport drug molecules or molecular imaging contrast agents into biological fluids and tissues and deliver or release them to specific targets. Incorporation into the nanocarriers aids the solubilization of hydrophobic drugs, improves the pharmacokinetic profile, regulates sustained release and prolonged blood circulation time when the therapeutic nanoagents are properly modified (e.g., by poly(ethylene glycol)-chains, PEG) to prevent immune clearance [9,10].

Surface modification with specific receptors has also been widely explored for targeted (active) delivery. Nevertheless, the concept of (passive) NP transport in the tumor blood vessel through inter-endothelial gaps has been considered for long time a characteristic feature of nanosized materials for nanomedicine [11]. It is worth noticing that, despite the fact that the development of NP to treat solid tumors is well consolidated, since due to their controlled size they are supposed to extravasate and access the tumor microenvironment, recently, Sindhwani et al. reported that indeed the transport of NP into solid tumors does not occur passively through the inter-endothelial gaps but, on the contrary, NP enter tumors via an active pathway that involves endothelial cells [12]. Despite the fact that the actual mechanism of action and uptake of NP is still debated and actively investigated, most scientists agree in considering NP potential toxicity as a major issue of the their application [6,13]. As a consequence, the development of biocompatible, biodegradable, bioresorbable and efficient versatile nanoplatforms for nanomedicine is in high demand and NP are requested to guarantee the safety of the treatment, minimizing simultaneously the side effects, and amplifying the therapeutic outcomes.

Some inorganic NP, for example, have been reported to accumulate largely in the reticuloendothelial system producing low passive targeting specificity and long-term toxicity and these results are expected to hinder the advancements of such nanomaterials for clinical trials [13]. In this context biomolecules deriving from living organisms, like melanin, are, on the contrary, gaining more and more attention, due to their native biocompatibility, their biodegradability, but also considering the multi-functional and various physiological roles they play in living organisms. The definition of melanin and melanogenesis was discussed in detail, for example, by d’Ischia et al. [14]. Melanins are pigments of diverse structure and origin resulting from the enzymatic oxidation and polymerization either of tyrosine in animals or of phenolic derivatives in other organisms. In humans, in particular, hair, skin, and eye color result from these pigments that are produced by specific cells: the melanocytes [14,15,16].

Nevertheless, not all the materials of biological origin can be considered totally safe especially in the form of NP. The possible toxicity of NP, in fact, is not simply related to the nature of the constituting materials but also to features like e.g., size, shape and surface properties. Hence materials considered safe can become hazardous at the nanoscale [17,18]. This nanotoxicity is in part related with the fact that many nanoagents are not discharged from the body, but on the contrary, they accumulate in vital organs jeopardizing their functioning [19]. Peng et al., for example, recently demonstrated, in animal models, that un-properly designed NP enhance both intravasation and extravasation of breast cancer cells, with the dramatic effect of extending existing metastasis and favoring the formation of new metastatic sites [20]. In this context, it is important to consider that, differently from other biomolecules melanin, is already present in the human body as nanosized particles in the melanosomes, organelles produced by the melanocyte and responsible for coloration and photoprotection in living organisms [21]. Hence, melanin is known to be biocompatible already in the form of melanin nanoparticles (MNP). Nevertheless we would like to stress that melanosomes do not contain only pigments but they are complex organelles that, for example, harbour numerous tissue-specific proteins [15,16].

This native biocompatibility of natural melanin, and in particular of eumelanin, that has been described in many studies as one of the main advantages of naturally extracted melanin, is fortunately maintained in its artificial bio-inspired copies like NP obtained by controlled oxidation and polymerization of dopamine and known as poly-dopamine (PDA) [22,23]. Various in vitro studies in different human-deriving cells have shown, in fact, that PDA as well as many complexes MNP exhibit minor to low cytotoxicity. Moreover, thanks to the versatility and simplicity of the preparative process PDA and related materials became, as summarized in Figure 1, unique building blocks for the design of highly biocompatible multi-functional theranostic nanoagents that would be difficultly achievable employing natural melanin sources.

Considering multi-functionality, an ideal biomaterial for nanomedicine has to join intrinsic biocompatibility to additional features [24] and melanin and its artificial analogous like PDA possess numerous fascinating properties [25,26,27,28,29,30]:(1)Broadband UV-Vis absorption. Melanins present interesting optical properties and in particular eumelanin but also PDA show a broad monotonic absorption band in the UV-Vis spectrum up to near-infrared (NIR) [6,31]. Although the origin of this typical absorption band is still debated, and objective of recent investigation based on ultrafast transient absorption (UF TA) techniques, the broad optical absorption is at the basis of the natural function of melanosomes that is imparting coloration and protecting from the potential dangerous effect of the solar radiation. The photo-protecting action is strongly related to the photophysical properties of melanin derivatives that absorb light very efficiently, producing excited states that deactivate mostly through ultra-fast non-radiative processes, without the production of any long-lived, potentially reactive transient species. As a consequence, eumelanin is photochemically very poorly reactive and it transforms light energy into heat. If the intensity of input radiation is moderate, like in the case of sunlight exposure, the heat produced can be dissipated by biological tissues safely, without causing significant local temperature increase [32]. On the contrary, upon locally controlled and intense light stimulation, the heat generation can be exploited for producing acoustic signals for diagnostic as in PAI or even induce a local hyperthermia and kill cancer cells as in PTT [33].(2)Antioxidant and free radical scavenging activity. Melanin as well as artificial MNP, possessing various reductive functional groups as catechol, amine, and imine, exhibit a broad action of defense against multiple reactive oxygen and nitrogen species (RONS), including O_2_^•−^, H_2_O_2_, ^•^OH, ^•^NO, and ONOO^−^, that are generated, e.g., after solar exposure, as well as in diseases [34]. Radical-enriched artificial MNP have been even reported to be so efficient to work even as ionizing radiation-protector able to scavenge reactive oxygen species produced upon X-ray exposure [35].(3)Metal-Chelation properties. Both natural and synthetic melanin possess multiple functional groups with binding sites for metal ions as iron, gadolinium, copper and manganese that can be exploited as contrast agents for MRI also in imaging-guided therapy [36,37,38,39,40].(4)Drug loading capacity. The various π-conjugated structures, such as dihydroxyindole, indolequinone (see Figure 2), on the surface of melanin make it possible the binding with various aromatic structures via π–π stacking or other secondary interactions as hydrogen bonding and wan der Waals interactions. As an alternative, drugs can be also either linked through covalent bonds to the melanin surface or they can be physically encapsulated within the polymer matrix [41,42].(5)Easy functionalization. Also, in this case, the presence of numerous functional groups as catechol, o-quinone, amine, and imine, endows melanin with reactive sites for the formation of covalent bonds with various reactants and in particular with ligands for target recognition or “stealth”-polymer like PEG [9].(6)Paramagnetism. Due to the presence of stable π-electron free radical species, paramagnetism is a characteristic feature naturally occurring in melanin, giving persistent electron spin resonance (ESR or EPR). Similarly, PDA exhibits a single-line ESR spectrum [35,43].

Thanks to this combination of features MNP are not simply carriers for the delivery of therapeutic or imaging agents but are themselves suitable to generate diagnostic signals, as in the case of PAI or FI or to perform dark or light stimulated therapeutic action as in the case of antioxidant (AOX) therapy and PTT respectively. MNP are hence themselves active materials in nanomedicine different from other NP that just constitute scaffold structures. The functionality of MNP can be further implemented by incorporating additional units like drug molecules, fluorophores, photosensitizers (PS) for PDT, and so on. In this case the versatility of functionalization of MNP is extremely advantageous allowing to exploit both covalent and non-covalent interaction or physical encapsulation. In this review article, we will discuss the most recent examples of application of MNP to nanomedicine focusing on the multiplicity of functions that melanin performs in theranostic platform, considering systems with increasing complexity. We will start analyzing more in the detail the main properties of melanin and its derivative as well as main natural sources and synthetic methods moving to application from mono-modal to multi-modal imaging and then to therapeutics and finally to theranostics and synergistic therapies including gene- and immuno- therapy in combination to photothermal and photodynamic. Melanin-related nanomaterials have been attracting more and more attention in the scientific community during last years, differently from other review papers, this article will focus on NP containing melanin analogues rather than nanostructures in general [44,45,46,47]. Considering that melanin in nature performs functions mimicked by some cosmetic formulations, like protecting and coloring, examples of possible applications of artificial MNP in cosmetics are increasing. Thus, we will underline the new interconnection between nanomedicine and cosmetics paving the way to the birth of a new field of science that we referred to as nanocosmetics. As a general approach, we will concentrate the critical discussion of the state of art of the research in the field of MNP in last section: conclusion and perspective. In this last part of the review, we will summarize and discuss the main limits of MNP for application in nanomedicine and nanocosmetics.

### 1.1. Melanin Types, Structure and Sources

#### 1.1.1. Melanin Types and Structures

Melanin is an abundant and heterogeneous biopigment found ubiquitously in many living organisms. The different types of the pigment can be classified based on their source as animal, plant, fungal, or bacterial melanin or based on their chemical and physical features as eumelanin, pheomelanin, neuromelanin, allomelanin, and pyomelanin [48]. Due to their rich diversity in terms of origin, size, color, and function, melanins are academically defined as biopolymers formed from phenolic compounds by polymerization via quinones. Animal melanin includes only eumelanin, pheomelanin, and neuromelanin, their biosynthesis is schematized in Figure 2. In nature, brown-black eumelanin and red-yellow pheomelanin are derived from tyrosine precursor. More in detail, natural eumelanin is biosynthesized from the tyrosinase-catalyzed oxidation of tyrosine or 3,4-Dihydroxy-L-phenylalanine (L-DOPA) to DOPAquinone [49]. Following, DOPAquinone is converted to L-Dopachrome through auto-oxidation, which tautomerizes to form 5,6-dihydroxyindole-2-carboxylic acid (DHICA) or reaction proceeds spontaneously with decarboxylation to give mainly 5,6- dihydroxyindole (DHI). Finally, eumelanin is formed through a series of oxidative polymerization reactions from DHI and DHICA, which are the main building blocks of eumelanin [50]. In the case of pheomelanin the biosynthesis involves the binding of cysteine in the positions 2 or 5 of the benzene ring, forming 5-S- and 2-S-cysteinyldopa from DOPAquinone. Subsequently, pheomelanin is formed via the oxidation of the substrates to cysteinyldopa-quinones, that cyclize in 1,4 benzothiazine intermediates [51]. Even if the supramolecular structures of the two biopigments remain still indefinite, eumelanin can be considered a heterogeneous polymer consisting of DHI and DHICA in various ratios, while pheomelanin is composed from the oxidative polymerization of 5-S- and 2-S-cysteinyldopa and consists of benzothiazine and benzothiazole units. Something that must be highlighted is that the two pigments do not only differ in the chemical structure but also in their physicochemical properties. Eumelanin is photoprotective, absorbing the UV-Vis radiation and dissipating the energy harmlessly, while pheomelanin is more photochemically active and is considered to be phototoxic [52]. For example, Premi et al. compared the induction of cyclobutane pyrimidine dimers (CPDs), known to be responsible of melanoma, in two populations of transgenic mice, one having melanocytes that synthesize red-yellow pheomelanin and, the other one, eumelanin. The experimental results demonstrated that pheomelanin not only was poorly effective in protecting from CPD formation but it also induced dark CPD generation [53]. Noonan et al. reported that indeed melanoma induction in the UVA is effective only in the presence of melanin pigment [54]. Nevertheless, the generation of CPD is slow and it can be efficiently blocked by the presence of antioxidant agents [53].

Neuromelanin is a dark insoluble pigment with complex composition that is closely related to the two other melanins, since it contains both eumelanin and pheomelanin segments. As discussed in detail by Zucca et al., neuromelanin is mainly produced in specific neurons of the substantia nigra and, in humans, it is mostly found in the catecholaminergic nuclei of the human brainstem. From the chemical point of view, it is produced by oxidation of catecholamines but its formation also involves bonding to other cellular components such as proteins, lipids, and metals [55,56]. Last but not least, allomelanin and pyomelanin are mainly found in plants and microorganisms. Pyomelanin presents a bacterial biopolymer and originates from the catabolism of tyrosine or phenylalanine through the auto-oxidation and self-polymerization of homogentisic acid (HGA) [57]. Allomelanin is a sulfur and nitrogen-free type of melanin mainly found in fungi. This pigment derives from nitrogen-free precursors as catechol and 1,8-dihydroxynaphthalene (1,8-DHN) [58]. Despite their diversity, all melanins possess fascinating physicochemical properties and play significant physiological roles.

#### 1.1.2. Natural versus Synthetic Sources

Melanin can either be naturally obtained, or synthesized using different approaches that will be analyzed in the following sections. Eumelanin and pheomelanin are present in different organs of both mammals and nonmammals including the hair, skin, eyes and feathers. Moreover they can be found in the ink of cuttlefish, squid, and other cephalopods, but also in microorganisms as bacteria and additional species. Due to the fact that the isolation of natural melanin from different mammalian tissues, as hair, requires harsh chemical conditions or processes that lack of a high extraction yield, melanin extraction from the ink sac of the cuttlefish, also known as sepiamelanin, is the most popular technique used. In recent years, sepiomelanin, mainly composed by eumelanin, has been utilized as the standard melanin material. The extraction method includes centrifugation with water or washing with salts [49]. In two different studies, melanin has also been successfully extracted from black sesame seeds and human hair [59,60,61]. However, it is obvious that the acquisition of melanin from different biological sources has the disadvantage of inherent variations in the physicochemical properties of the biopigments, something that increases the need of a standardized extraction and purification method. On the other hand, PDA is also considered to be the synthetic analogue of eumelanin, even if the synthetic pathway of PDA differs from the one of natural melanin. PDA displays optical, electronic and magnetic properties identical to those of natural eumelanin being characterized also by excellent biocompatibility. Additionally, PDA owns many functional groups, such as catechol, amine, and imine, that can either covalently bind or non-covalently load through π-π stacking, hydrogen bonding, van der Waals forces, etc., and different active agents. With these benefits, PDA can be used for a wide range of application acting as coating material, as carrier or as the active component for bio-application [23].

### 1.2. Preparation Methods, and Polymerization Mechanisms

#### 1.2.1. Preparation Methods

MNP preparation is usually achieved via one of three main approaches: (i) solution oxidation, (ii) enzymatic oxidation and (iii) electro-polymerization [23,47]. Solution oxidation method is widely used because of its simplicity. Dopamine oxidation and following self-polymerization take place spontaneously under mild alkaline condition with solvated oxygen. Reaction evolution could be tracked monitoring solution color change (from colorless to deep brown/black). This method could be employed also for the preparation of PDA films or NPs coatings [23].

Enzymatic oxidation is a complementary method with some interesting features. In general, enzymatic catalysis has been widely exploited in various industrial processes, because of its efficiency and since this method is environmentally friendly. Biosynthesis of melanin itself in organisms is carried out via catalytic oxidation of l-tyrosine by tyrosinase, for this reason PDA produced by enzymatic method tends to be more similar to natural melanin [23]. The enzyme laccase, a multi Cu-containing polyphenol oxidase, has been employed for a wide range of applications, such as water remediation, textile bleaching and microbial transformation of natural products [62]. More importantly, dopamine oxidation and polymerization could be also achieved via laccase action at pH 6 [62]. The enzyme-catalyzed strategy possesses relatively complicated processes but higher efficiency without waste production [63].

Finally, in electro-polymerization process PDA is produced by direct oxidation and deposition on the electrode. The greatest advantage of this method, beyond simplicity, is the fact that, by adjusting a wide range of parameters, (e.g., pH, temperature, monomer concentration, etc.), various types of nanostructure could be achieved [22].

#### 1.2.2. Polymerization Mechanism

Polymerization mechanism of PDA has been still disputed in scientific community, anyway it is believed that the involved stages could resemble the natural melanin synthetic pathway [23]. First step consists in dopamine oxidation to dopamine-quinone followed by 1,4 Michael-type intramolecular cyclization to leucodopaminechrome [64]. Then, different isomers of 5,6-indolequinone are obtained via oxidations and rearrangements, increasing the system complexity. At this point, different mechanisms are suggested. However, little experimental evidence has been reported to date. On the basis of FTIR analyses, the resulted comparison between PDA and dopamine spectra seems to verify the indole formation via Michael addition [65]. Bielawski et al. confirmed this result via solid state ^15^N-NMR spectroscopy, were they observed the disappearance of the peaks related to NH_2_ scissoring vibration and CH_2_ bending. Moreover, on the basis of one-dimensional solid state ^13^C-NMR, they claimed that polymeric chain is composed by saturated indoline structures, and that PDA was mainly composed by an aggregation of monomers crosslinked by strong non-covalent bonding [66]. Finally, Lee and co-workers suggested a hybrid mechanism in which PDA is composed by the coexistence of supramolecular structures in which unpolymerized dopamine was self-assembled with its main oxidative products and covalent-bonded oxidative polymerization products [67].

### 1.3. Physicochemical Properties

#### 1.3.1. Photophysical and Photochemical Properties of MNP

Interaction of melanin with light is at the basis of its function in Nature and of most of MNP bio-application. Both natural eumelanin and synthetic MNP are optical absorbers with the properties of an amorphous semiconductor. They both exhibit broadband monotonic absorbance in all the UV-Visible range up to near-infrared (NIR), which reminds more of an inorganic semiconductor material with a small optical band gap (about 0.5 eV) [68] rather than an organic chromophore with specific absorption peaks typically associated to transitions from bonding to antibonding π orbitals. On the other hand, broad band absorption also results from suitable combination of sets of chromophores with different size and number of unsaturated bonds since the energy of the transitions from bonding to antibonding orbitals (π → π*) is decreased by conjugation, especially in the aromatic rings, with a concomitant shift of the absorption band from the UV to the NIR [6]. Indeed, most recent investigation of the electronic properties of synthetic melanin based on ultrafast transient absorption (UF-TA) spectroscopy revealed the presence of a subset of chemically heterogeneous chromophores rather than of a homogenous semiconductor material with a defined band-gap (Figure 3) [69]. Surprisingly, despite the coexistence of different chromophores, the constancy of the photoinduced absorption band independently on the excitation wavelengths (in the 265–600 nm range), and its common decay kinetics, show that all the different chromophores present in eumelanin, independently on their actual chemical composition, deactivate following a single, common electronic pathway. According to Kohl et al., this process involves the generation, in less than 200 fs after the excitation, of charge transfer excitons localized among graphene-like chromophores. The presence of chromophores with a typical structure arising from the combination of sp^2^ -hybridized carbon and nitrogen atoms was indeed confirmed by Raman spectroscopy. Interestingly, this study spreads new light on the structure of melanin suggesting that the self-assembly of graphene fragments and molecular components of eumelanin are likely to be similar [70,71]. The effect of the supramolecular organization on the photophysical properties of MNP was evidenced also by Ju et al. who investigated the nature of the broad-band absorption of eumelanin, concluding that the size-dependent absorption is associated with the geometric packing order of eumelanin proto-molecules in the large-scale aggregation. According to this study larger PDA nanoparticles showed a higher attenuation at longer wavelengths compared to smaller particles due to their different aggregation structure. Transient absorption spectroscopy was used also to provide more details about the effect of aggregation on the broad absorption band of MNP [72] and results confirmed the hypothesis that broad absorption bands governed by intrinsic π-electron delocalization within fundamental eumelanin oligomers can be further affected by secondary interactions such as π−π stacking and aggregation in the hierarchical assembly system of eumelanin [73]. Despite these results demonstrate the supramolecular properties of melanin, the presence of quite extended polymeric sections in PDA was demonstrated by Messersmith and coworkers using single-molecule force spectroscopy (SMFS) [74]. In particular a cantilever was functionalized with PDA and approach/retract experiments to a bare oxide surface revealed the characteristic traces of up to 200 nm long polymeric chains. All these partially contradicting results demonstrate that the structure of melanin and MNP needs further elucidation.

In order to understand the main mechanism of excited state deactivation in MNP it is essential to recall that these materials typically show a very low fluorescence quantum yield (QY < 0.1%) [75,76]. Considered the fast decay observed in the UF TA experiments, that also showed no evidence of the formation of any long living species, it can be concluded that MNP excited state deactivates almost quantitatively via a non-radiative path with a rate constant that can be estimated to be as high as ~10^13^ s^−1^. Ultra-short excited state lifetime explains the high photo-stability and poor photochemical reactivity of melanin-based bio-pigments.

The combination of these features with a broad absorption band increasing toward high energies allows an efficient absorption of the UV− vis light and its transformation into heat without the generation of any reactive intermediate, which is beneficial for photoprotection. These absorbing properties are the basis of melanin protective action in low to moderate irradiation intensity conditions (like in sunlight exposure) in which heat can be dissipated by biological tissues without producing significant local hyperthermia [77]. On the contrary, short and intense laser light pulses can be used to produce fast local release of thermal energy from MNP either to achieve selective and controlled cell death (as in PTT) or to induce a pressure wave that can be detected as a diagnostic signal (as in PAI). It is important to stress that, thanks to their broad absorption signature, MNP can be excited in the NIR region, which is more suitable for in vivo application since tissues are more transparent than in the UV-Vis. Additionally technology of high-power, short-pulse laser is more advanced for NIR range and several commercial and versatile sources are available even for low-cost applications. Additionally, the photothermal response of MNP can be in part controlled by tuning the NP size [73]. Finally, photoinduced heating can be exploited to enhance the release of therapeutic agent loaded onto the MNP when used as drug delivery scaffolds.

#### 1.3.2. Antioxidant (AOX) Activity

Reactive oxygen and nitrogen species (RONS) are involved in fundamental physiological processes as signaling units; nevertheless, their excessive formation can contribute to the development of pathological conditions. This duality, discussed in detail by Schieber and Chandel [78], emerges in the case of oxidative stress, a condition in which excessive cellular concentration of reactive oxygen species (ROS) starts to damage lipids, proteins and DNA. Oxidative stress is responsible for many biological processes including photoaging and it is involved in several important pathologies [79,80]. The enzymatic or non-enzymatic defense mechanisms are often unable to counteract the excess of ROS leading to their accumulation in cells. Hence a huge variety of products intended to restore this balance, containing different kind of antioxidants (AOX), appeared on the market even in the form of dietary supplements [81]. Indeed, the natural photo-protective role of eumelanin is not simply based on shielding from UV/radiation, but it exploits its ability to scavenge all kinds of free-radical that are produced by UV/radiation [82]. MNP, in fact, offer a broad defense against multiple ROS, including O_2_^•−^, H_2_O_2_, ^•^OH and NO^•^, that are mainly generated in diseases [34]. Thus, the multi-antioxidative activity of the bioinspired MNP has been explored for AOX therapy. These NP are so efficient that recently, novel PDA NP have been reported to provide protection against ionizing radiation-produced radical [35]. This was achieved by introducing stable nitrogen radical in synthetic PDA NP an approach that significantly enhanced the X-ray protection properties of the MNP, also providing them novel electrochemical and magnetic properties. Importantly, these NP were successfully internalized by human epidermal keratinocytes, mimicking the behavior of natural melanosome, where they significantly scavenged X-ray-induced ROS (Figure 4) [35]. Further examples of the application of MNP to AOX therapy will discussed in this review. In an alternative strategy, the AOX properties of MNP were also exploited to stabilize easily degradable, oxygen sensitive molecules, suitable as promising but unstable drugs that, when incorporated in MNP, showed enhanced activity.

#### 1.3.3. Metal Ion Chelating Ability and Redox Activities

Many functional groups in MNP, (*o*-quinone, carboxy, amino, imine and phenols) are capable of an effective metal ions binding [83]. For this reason, PDA could be successfully used for complexing several multivalent metal ions such as Fe(III), Cu(II), Zn(II), etc. [23]. Moreover, the binding sites of PDA are activated at different pH, resulting in an interesting pH-dependent chelating system [84]. As will be shown, the formation of Gd^3+^ chelates which increase the signal intensity on T_1_-weighted images in MRI is particularly important for nanomedicine [40].

In addition to chelation ability, it is worth mentioning the ability of PDA to reduce noble metal ions such as Ag(I), Au(III) and Pt(III) in alkaline conditions, which can be exploited for the deposition of PDA films on metals [23].

#### 1.3.4. Functionalization and Coating: Use as a Bio-Template or as a Coating

Melanin bioinspired materials have found extensive application in material science for their strong adhesive properties mimicking the strategy used by several marine organisms. As discussed by Ruiz-Molina and coworkers, adhesion results from different kind of catechol-surface interactions including hydrogen bonding, π- π stacking, coordination and covalent bonding by reaction with surface groups [85]. Nucleophilic molecules with thiol- amino-groups result to be ideal for this scope. For example, the quinone groups resulted from catechol oxidation, can react with amines via a Schiff base reaction. Similarly, Michael addition reactions could be used, especially with thiols [86]. These reactions prove to be very simple, and they can be carried out in mild condition and simple apparatus. It is worth to mention that these reactions can proceed in water solution, and that the coupling products remain quite stable in time [87]. Since adhesion to surface involves the catechol, Ruiz-Molina and coworkers proposed also the use of PDA analogous demonstrating that some bis-catecholic polymers can be used to coat mesoporous silica NP in order to slow down the release of preloaded model molecules [88]. Recently, Messersmith and coworkers demonstrated that, indeed, having both catechols and amines combined in the same functional monomer is advantageous for the final adhesion strength: decoupling the two kinds of functional groups, in fact, prevents cooperative effects with an observable worsening of the adhesive performances [89]. The same author also showed that mechanical properties of PDA coating can be further enhanced, and the roughness reduced, by treatment with blue-diode laser treatment, inducing graphitization while preserving multifunctionality [90]. Surface adhesion of PDA and related materials have been investigated in detail also by Lee and coworkers [91], who demonstrated that the stability of the obtained coatings is widely independent on the chemical composition of the surface itself [92]. This feature is surely advantageous since it allows a very general approach to surface functionalization. Nevertheless, as recently demonstrated by the same authors, material selectivity becomes preferable for the differential coating of a multi-material structure [93].

Application of the adhesion properties of melanin inspired material for coating has been discussed in detail in a recent review article and it is to a large extent out of the scope of this work, which is focused on MNP [22,94].

## 2. MNP for Bio-Imaging

A key concept of nanomedicine is multi-functionality and most recently developed platforms for nanomedicine combine diagnostic, possibly multimodal, imaging and therapeutic action in an integrative theranostic approach [1,2]. In this framework, intrinsically multifunctional materials like MNP present a unique versatility that constitutes one of their main features that is widely explored in most advanced applications. Considering that the most active research in the field is devoted to designing new systems that exploit this multi-functionality, in many cases it is not possible to discuss separately the diagnostic and therapeutic properties of complex system or even to de-couple different imaging modes. This is, for example, due to the fact that the same stimulus used for interrogating the system for diagnostic purposes produces multiple signals and may simultaneously perform a therapeutic action. This is particularly evident in the case of pulsed laser light stimulation that may induce fluorescence but also PA signal while the heat released may produce local hyperthermia with a therapeutic action (PTT). Despite this partial limitation, we preferred to discuss the application to imaging first in this section, focusing on the different techniques, and then to consider theranostic in the next section with the purpose of increasing gradually the complexity of the described of systems for the sake of clarity.

### 2.1. Fluorescence Imaging (FI)

So far, fluorescent NP have attracted a lot of interest because of their superior optical properties and photostability compared to fluorescent molecular dyes [95,96,97]. Since some fluorescent inorganic NP are toxic and not degradable [98], in the last years, fluorescent organic nanoparticles (FON) have emerged for biomedical applications [99,100]. Melanin-based FON have been developed for bioimaging because of their remarkable biocompatibility and, in part, for their photophysical properties [101,102].

Shi et al., for example, reported a novel polymer-FON conjugate that have been fabricated via combination of reversible addition-fragmentation chain transfer (RAFT) polymerization and self-polymerization of dopamine and polyethyleneimine (PEI) [103]. One of the most important features of this synthetic procedure was that the chemical compositions and functional groups of the formed FONs could be well adjusted through RAFT polymerization, giving the possibility for the construction of a variety of a set of different polymer-FON conjugates. Even if the aforementioned FONs possessed good biocompatibility and strong fluorescence, other important parameters such as the size and the morphology could not be easily tuned by this synthetic approach. Exploiting the extraordinary fluorescence properties and good biocompatibility of carbonaceous dots, Xiao et al. suggested a one-step hydrothermal method for the synthesis of melanin carbonaceous dots (MCDs) [104]. Due to their excitation and emission peaks in the red-NIR region, the synthetized MCDs were found to be ideal for in vivo fluorescence imaging as a deeper tissue penetration could be achieved with reduced background. Regarding tumor regions, MCDs accumulated efficiently in 4T1 tumor site giving a strong fluorescence signal 24 h after the injection. Despite the well documented biocompatibility of melanin, in this case, the overall toxicity of the imaging agent was not negligible.

Fluorescence of MNP is surely an interesting and potentially useful property but it presents some limitations: fluorescence peaks of MNP are usually broad, interfering with the simultaneous use of other fluorophores [105] and the fluorescence features are not always completely controllable in these kind of systems. Additionally, fluorescence quantum yield of MNP is typically low. The intrinsic fluorescence properties of MNP have been recently reviewed by [75] and their application for bioimaging becomes exceptionally valuable, mostly in combination with other techniques such as X-ray computer tomography (CT) in a system proposed by Zhang et al., who developed a MNP-based probe for tumor-targeted dual mode imaging guided PTT [106]. On the other hand, more effective fluorescent MNP can be achieved upon the incorporation of specific fuorophores. An interesting example was given by Wang et al. that developed a fluorometric method for glutathione (GSH) detection in biological fluids [107]. They synthesized near-infrared fluorescent polymer dots (p-dots) from poly (stryrene-co-maleic anhydride), poly[(9,9′-dioctyl-2,7-divinylenefluorenylene)-alt-2-methoxy-5-(2-ethyl-hexyloxy)-1,4-phenylene] and tetraphenylporphyrin, that were functionalized with dopamine. In a weak alkaline environment, polymerization occurs and a polydopamine coating is formed on the p-dot, quenching its NIR fluorescence (656 nm). In the presence of GSH, dopamine self-polymerization was inhibited, preventing quenching. An antiquenching effect study was performed and a GSH detection limit of 0.06 μM was calculated. Moreover, the selectivity of the system was measured over some potential non-targeted interferences, such as some metal ions, proteins, amino acids and glucose. The experimental data demonstrated an excellent selectivity of the melanin-p-dots for GSH detection. Dopamine p-dots were also tested for intracellular GSH detection and in human serum samples, showing good selectivity and sensitivity, so demonstrating the applicability of this system in real environment.

In another example Wu et al. exploited the fluorescence of green fluorescence protein (GFP) to achieve a multifunctional theranostic agent that was based on mesoporous polydopamine nanoparticles with a plasmonic core and was able to perform functional protein delivery with fluorescence real time monitoring [108]. These authors synthesised core-shell mPDA coated Au nanoparticles by promoting dopamine self-polymerization in a seed-mediated microemulsion method, using Pluronic F-127 and 1,3,5-trimethylbenzene as pore template and pore swelling agents respectively. The loading of two polyhistidine-tagged proteins (green fluorescent protein, GFP and ribonuclease A protein, RNase) in the pores could be favoured by metal ions (i.e., Ni^2+^), complexed by the catechol groups of polydopamine. As shown in Figure 5 the proteins could be released into the cell by means of NIR light irradiation, which destabilised the coordination bonds of polydopamine and metal ions. This stimuli-responsive protein release, combined with the passive accumulation of the NP in tumour regions, successfully inhibited tumour growth in vivo. The loading with green fluorescence protein (GFP) permitted a real time release monitoring. It is worth noticing that in this system the Au plasmonic core plays a double role since it also acts as a quencher of the fluorescence of GFP, as a consequence no fluorescence signal can be detected as far as the protein is inside the nanostructure. On the other hand, once NIR irradiation induces GFP release, fluorescence is turned on and a significant signal can be detected.

### 2.2. Photoacoustic Imaging (PAI)

Photoacoustic imaging (PAI) technique is based on photoinduced thermal response [109]. Light absorption from molecules produces a thermally induced pressure jump, which lead to ultrasonic waves signal, read by acoustic detectors [110]. Melanin is an excellent candidate for this kind of imaging because non-radiative deactivation is favoured after light irradiation, leading to heat release and because the signal can be generated by NIR light irradiation, which allows higher imaging depth [111]. Considered these features, Fan et al. verified the native photoacoustic properties of PEG-melanin nanoparticles (PEG-MNP) [43] and they demonstrated a detection limit as low as 0.625μΜ in aqueous solution. A good linear dependency of the signal on the NP concentration was also reported in vitro. For the in vivo experiments and in particular for PA melanoma imaging, MNP structure was modified in order to enhance the sensitivity by accumulation via targeted recognition. PEG-MNP were, in fact, modified with cyclic Arg-Gly-Asp-D-phe-Cys peptide (RGD), a ligand able to target tumour α_v_β_3_ integrin and achieve U87MG tumour accumulation. Indeed, after 4 h of the injection of RGD-PEG-MNP in mice, an obvious increase in the PA tumour signal was observed in comparison to the signal observed after the injection with simple PEG-MNP. Following a different synthetic approach, Repenko et al. fabricated a controlled covalent form of PDA for PA imaging via nickel-catalysed Kumada cross-coupling [112]. After the polymerization process, PDA could be oxidized into a linear form of melanin, leading to the desired water-soluble melanin-based polymer. The formed melanin polymer exhibited high PA contrast properties with a detection depth of more than 10 mm. These features, accompanied with the minor bio-toxicity of these MNP, are very promising in view of the general application of this novel material for PAI. In order to amplify the intensity of the PA signal of MNP, Ju et al. proposed the functionalization of their surface with hydrolysis-susceptible citraconic amide [113]. The resulting pH-responsive MNP were able to aggregate spontaneously in mildly acidic conditions. This process did not change substantially the optical absorbance but it increased noticeably the PA signal intensity in the NIR window of biological tissue when compared to the signal in neutral environment. From the mechanistic point of view, amplification was explained as resulting by the overlapping of thermal fields of the aggregated MNP. Interestingly, this stimuli-responsive PA signal amplification mechanism is, at least in principle, suitable for monitoring and detecting the acidic tumor microenvironment. Focusing on the diagnosis of articular cartilage degeneration in osteoarthritis, Chen et al. developed a positively charged melanin-based PA contrast agents [114]. More in details, these cationic systems were obtained by the functionalization of MNP with poly-L-lysine and they were able to strongly interact with the anionic glycosaminoglycans in the cartilage. Particularity, possessing a zeta potential between +32.5 ± 9.3 mV the formed nanoparticles exhibited an enhanced cartilage uptake, longer retention time as well as high PA intensity, which was almost two folds stronger in healthy joints than in osteoarthritis joints. The last feature was essential for the early detection of the disease in vivo.

Although this section is specifically dedicated to PAI, the most recent developments of MNP design for nanomedicine aim to achieve multifunctional therapeutic agents rather than simple imaging agents. In particular PAI can be, in general, difficulty considered separately form photo-thermal therapy (PTT) since both PAI and PTT exploit a common property of MNP, namely the ability to convert absorbed NIR radiation into heat very fast (picosecond time scale) and efficiently (about 100% efficiency). Therefore MNP are intrinsically theranostic agents that combine both PAI and PTT and their multi-functionality can be implemented by further functionalization as, for example in the following system reported by Zhang et al. [115]. These authors proposed a melanin-based theranostic agent (LysoIr@PDA-CD-RGD) resulting from the modification of a drug delivery nano-system with a phosphorescent iridium (III) complex (LysoIr) that presents efficient two-photon absorption and anticancer properties. Thanks to the functionalisation with arginine-glycine-aspartic acid tripeptides (RGD), the system was able to deliver LysoIr to integrin-rich tumour cells, combining chemotherapy and two-photon absorption phosphorescence lifetime imaging microscopy (TP-PLIM). The PDA matrix was, on the other hand, exploited for combined PTT/PAI. In vivo photoacoustic imaging tomography (PAT) was performed after injecting the nano-system in mice bearing U87 tumours and irradiating with an 810-nm laser. A strong photoacoustic signal was detected in the tumour sites, confirming LysoIr@PDA-CD-RGD as an effective theranostic anticancer system. A more sophisticated and informative PAI technique was exploited by Gujrati et al. who demonstrated the use of multi-spectral optoacoustic tomography (MSOT) for MNP [116].

These authors encapsulated melanin in bacterial outer membrane vesicles (OMV^Mel^) using a bacterial strain expressing tyrosinase transgene (Figure 6) and therefore, obtained 20–100 nm vesicles which were suitable for tumour detection. Tests performed on mice exhibited a high MSOT signal soon after intravenous injection, which increased up to 120 min, confirming the accumulation in the tumour by EPR effect.

### 2.3. Magnetic Resonance Imaging (MRI)

Magnetic resonance imaging (MRI) is one of the most common diagnostic tools because it is non-invasive and it offers an acceptable spatial and temporal resolution [38]. Many complexes of transition metal and lanthanide paramagnetic ions have been studied as contrast agents for MRI since they can decrease the nuclei relaxation times of neighbouring water protons, enhancing the image contrast [117]. So far, the most used contrast agents are Gd^3+^ chelates which increase the signal intensity on T_1_-weighted images [40]. Nevertheless, most of these complexes present low sensitivity, non-specificity, fast elimination from the circulation system and toxicity [118]. Taking these drawbacks into account, new families of contrast agents based on melanin have been recently developed, because of its biocompatibility and its strong metal-ions chelation ability [25].

Cia et al. proposed a simple example of Gd^3+^-chelated MNP for MR monitoring and tracking of bone mesenchymal stem cells in vivo [119]. The formed imaging agents possessed ultra-small size, around 7 nm, and were able to effectively label the cells of interest through endocytosis. Moreover, numerous advantages of the novel imaging agents were reported, including their exceptional stability and sensitivity. Thanks to the shorter T_1_ relaxation time and high cell labelling ability of the MNP based nano-probe, labelled stem cells could be detected even four weeks after their intramuscular injection. A more complicated system, a melanin–Gd^3+^ multifunctional system for MR imaging-guided therapy was presented by Zhan et al. [120], who developed an interesting theranostic antitumoral system in which adenine modified PDA (A-PDA) nanoparticles were bonded to a thymine-Zinc-Phthalocyanine complex (T-ZnPc), combining photothermic therapy (PTT) achieved by irradiation of PDA at 808 nm and photodynamic therapy (PDT) given by irradiation of the PS ZnPc at 665 nm. In this case Gd^3+^ ions were embedded in the MNP by simple mixing, exploiting the chelation of PDA by the catechol moieties. These NPs were able to efficiently accumulate in the tumour and to enhance significantly the MRI T_1_-weighted signal intensity. The relaxivity parameter (r_1_), which represents the dependence of the relaxation rate of the solvent to the concentration of the paramagnetic species [117], was 2.74 mM^−1^ s^−1^, similar to commercial MRI imaging agent Gadodiamide (3.47 mM^−1^ s^−1^). Unfortunately, it is well known that Gd^3+^ is harmful for patients with kidney injury and its accumulation in brain also in the case of people with normal renal functions was observed [121]. Therefore, the use of other paramagnetic ions as contrast agents for MRI have been explored. So far, Mn^2+^ has represented a valid alternative because it has physical properties similar to Gd^3+^ (i.e., high spin quantum number, long longitudinal electronic relaxation times and fast water exchange kinetics) but it is biogenic. Nevertheless, even if Mn^2+^ is less harmful than Gd^3+^ [122], its cellular toxicity is still not negligible [123]. Even if toxicity remains a challenge, the use of MNP, able to efficiently chelate Mn^2+^, is expected to lead to an enhancement in biosafety. In the light of this, Mn^2+^-chelated melanin nanoparticles were reported and, in particular, ultra-small MNP-PEG-Mn proposed by Xu et al., possessing a size of about 5.6 nm demonstrated the ability of deeper-penetrate tissues [37]. These water soluble MNP-PEG-Mn exhibited much higher r_1_ longitudinal relaxivity than existing T_1_ MR contrast agents and, therefore, an efficient and specific tumour-targeting MRI was achieved. Last but not least, even if the contrast agents showed a very high cancer cell-targeting specificity, their cytotoxicity was negligible as they were efficiently eliminated by both the renal and hepatobiliary pathways. Another interesting example of melanin-manganese composites for MRI was described by Addisu et al., who developed a biomimetic Alginate-dopamine and Alginate-polydopamine nanogels complexed with calcium/manganese ion (AlgDA(Ca/Mn) NG and AlgPDA (Ca/Mn) NG) [39]. In this approach, the presence of calcium and manganese ions conferred a high stability resulting from the complexation of calcium by alginate moieties and of manganese by the catechol groups of dopamine. These nanogels were highly biocompatible and also enhanced the relaxivity values r_1_ (longitudinal) and r_2_ (transverse). Illustratively, AlgPDA (Ca/Mn) NGs presented a three-fold higher relaxivity than commercial Gd-DTPA. In these nanogels water could easily diffuse thanks to catechol and carboxyl groups which bind the water molecules, favouring their interaction with the manganese core and that enhancing the contrast effect.

Considering other MRI contrast agents, iron-oxide based materials are commonly used to shorten T_2_ relaxation time, leading to a negative contrast [124]. A theranostic system based on superparamagnetic iron oxide (SPIO) and melanin was developed by Guan et al. who loaded mesoporous polydopamine nanoparticles (MPDA NPs) with ultra-small SPIO NPs and a drug (sorafenib, SRF), by means of π-π stacking and/or hydrophobic interaction [125]. The advantage of these novel design was that NIR irradiation of SPIO-SRF and melanin could be exploited to induced ferroptosis and PTT respectively, while, simultaneously, SPIO NP could act as negative contrast agent for MRI, shortening T_2_ relaxation time. In fact, SRF@MDPA SPIO were tested for MRI imaging both in vitro and in vivo and an effective increase of T_2_-weighted signal intensity was shown with the increase of Fe concentration, resulting in darkened images. In order to investigate the effect of the nature of the metal on the properties of melanin-based MRI agents, Chen et al. investigated the ultra-small melanin NPs complexed with several paramagnetic ions and compared their performances in contrast enhancement (Figure 7) [36]. Ultra-small water-soluble MNP were easily synthetized by dissolving melanin in an alkaline aqueous solution and sonicating at neutral pH. Afterwards, the synthetized melanin was modified with PEG in order to retain the water solubility, prolong the circulation in blood and thus improve the in vivo performance of the nanoagents. The affinity of melanin-PEG nanoparticles (MNP-PEG) for Gd^3+^, Mn^2+^, Fe^3+^ and Cu^2+^, depended on their ionic radius and ligand geometry, and it was reported to decreased following order Fe^3+^ > Cu^2+^ > Mn^2+^ > Gd^3+^. The authors demonstrated that all the nano-systems maintained ultra-small sizes (3–7 nm), high dispersity and high stability. Moreover, MRI tests, both in vitro and in vivo, proved the excellent contrast properties of the NP, probably improved by surface effects due to very small particle sizes. Gd^3+^ and Mn^2+^-doped nanoparticles gave the best outcomes in terms of relaxivity values. Moreover, the most intense signal was observed in liver and kidney after 1 h or 2 h for all the metal ions. Also, this result was imputed to the very small size of the NP which were efficiently cleared from renal and hepatobiliary pathways, suggesting their biosafety, a feature that makes them good candidates for kidney and liver MRI imaging.

### 2.4. Multimodal Imaging

Bioimaging technique with high sensitivity usually have a poor resolution, while the ones with a remarkable resolution lack in sensitivity; thus, multimodality can combine complementary abilities of each technique, powering imaging efficiency [126]. Also, for this reason, NP systems for multimodal imaging gained recently a lot of attention. MNP capability to bind both ions and molecules, makes them suitable for multimodal imaging applications. In most of the cases, the NP possessing dual-modal or multimodal imaging capacity also present therapeutic capabilities, e.g., as previously mentioned, PAI agents are generally suitable for PTT.

Wu et al. showed that MRI and PAI can be combined by embedding paramagnetic ions in MNP and exploiting their NIR absorbance and efficient thermal deactivation [127]. In particular Mn_2_(CO)_10_ was embedded in mesoporous PDA NP (MnCO@MPDA) to create a theranostic system able to perform CO release and PT cancer therapy in combination with MRI/PAI detection. Initially, MPDA were synthetized by using Pluronic F-127 and trimethylbenzene as pores template agents in order to obtain a 260.8 nm final NP size with pores of 8 nm. It was hence found that the loading of Mn_2_(CO)_10_ caused only a slight change in the size of the NP, which became 270.6 nm. Importantly, CO could be released from the system by the interaction with H_2_O_2_ highly present in the tumour environment. In addition, the simultaneous release of Mn^2+^ was reported to enhance the MRI contrast selectively in the intratumoral area while strong NIR absorption and photothermal conversion made MnCO@MPDA effective also as a PAI contrast agent. More in detail, cellular MRI/PAI tests revealed a signal increase for both imaging techniques up to 9 h after injection, whereas in vivo tests confirmed the effective tumour accumulation of the NP in mice. In alternative, Lemaster et al. reported a MRI/PAI multimodal system based on gadolinium-loaded synthetic melanin nanoparticles (Gd(III)-SMNPs). The incorporation of gadolinium in MNP was achieved by displacement method of Mn(III)-doped melanin-nanoparticles, exploiting the different affinity for the two metals. This strategy yielded higher Gd(III) loading than the most common direct post-modification method [40]. Interestingly, the resulting Gd(III)-SMNPs showed imaging performances similar to those of commercially available MRI contrast agents as Gd-DOTA and Gd-DTPA, but in addition they also presented a 40-fold increase of the PA signal when compared to metal free synthetic melanin alone. The PA signal of Gd(III) loaded MNP was even higher with respect to MNP doped with Ni(II), Zn(II), Cu(II), Mn(III) or Fe(III). The authors explained these different behaviours in terms of different optical absorption: metal doped SMNPs showed in fact a broad absorption band at 720–760 nm, probably due to metal coordination by catechol groups with the maximum intensity for Gd(III). Hence the higher absorbance let to higher PA signal. The properties of the Gd(III)-SMNPs were efficiently exploited in stem cells imaging, to demonstrate their actual bio-applicability [128]. Focusing on the combination of two other imaging modalities, Wang et al. presented a theranostic melanin-based system for PA and ultrasound (US) bimodal imaging [129]. Generally, ultrasound imaging is based on the generation of microbubbles as contrast agents and it very convenient for real-time cancer monitoring. However, this technique lacks in high resolution and contrast, drawbacks that can be compensated by photoacoustic imaging [130]. For combined US and PAI, core-shell melanin-poly (lactide-*co*-glycolic acid) (PLGA) nanoparticles were synthesized and loaded with paclitaxel (PTX) as anticancer drug and perfluoropentane (PFP). Melanin was used as photo-thermal agent that, upon irradiation, increases the local temperature up to the boiling point of the perfluorocarbon. After this, the nanoparticles are vaporized into microbubbles. The phase transformation of the nanoparticles into microbubbles of PFP, caused thermoelastic expansion, cavitation and activation of the nanoparticles. The last phenomena resulted not only in controlled PTX drug release but also, they directly damaged tumor tissues and promoted the penetration and absorption of the drug by increasing the local microvascular and cell membrane permeability. In a word, the melanin mediated photothermal effect in cooperation with the NIR-responsive drug release capability of the biosafe nanoparticles resulted in strong synergistic antitumor effect accompanied with accurate treatment guidance. Additionally, the role of melanin in these nanoparticles was also essential in order to promote both ultrasound and photoacoustic imaging. When this dual-modality imaging capacity of the nanoparticles was tested both in vitro and in vivo, good imaging performances were achieved both for US and PAI [131]. Moving towards other melanin-based multimodal imaging agents, PDA nanoparticles modified with lanthanides, which have interesting luminescence properties and a strong X-ray attenuation [132], can find application in the combination of fluorescence imaging and computer tomography (CT) techniques. For example, Zhang et al. reported Eu (III)-modified PDA NP for CT and photoluminescence (PL) dual-mode imaging accompanied with PTT [106]. PDA NP were synthesized by the self-assembly of dopamine hydrochloride in an alkaline environment and then modified on the surface by 3-chloropropionic acid and folic acid. Then a europium complex (Eu-(AA)_2_(DTA)Phen) was grafted on the particle surface by means of surface-initiated atom transfer radical polymerization (ATRP). In vivo trials showed relevant CT signal enhancement in the tumour region mainly at 24 h after intravenous injection. Regarding fluorescent imaging, an increase of the signal was reported in the first 8 h after injection in mice kidney and liver; the system was then metabolized almost completely after 48 h. Chen et al. proposed another fascinating system combining PDA upconversion NP (UCNP) and mesoporous silica to achieve core-shell NP labelled PDA@UCNP@mSiO_2_ NPs suitable for drug release and multi-mode detection by combining upconversion luminescence (UCL), CT and MR imaging. The PDA nanoparticles were produced by self-polymerization in alkaline conditions and, after the addition of PAA, Y^3+^, Yb^3+^, and Er^3+^ were adsorbed on the surface and then converted into rare earth hydroxides (RE(OH)_3_). Hence, the system was coated by a silica shell and a further modification with oleic acid and NaF led to the final structure. The presence of lanthanide ions made these nanoparticles suitable contrast agent not only for CT, but also for MRI. Furthermore, since lanthanide oxyfluorides have excellent up-conversion properties [133], UCL could be also generated. CT/MRI was applied to tumour-bearing mice, showing a high contrast imaging for the first 24 h after injection. The PDA@UCNP@mSiO_2_ NPs possess a strong green UCL under 980 nm irradiation a signal detectable also after incubation with HepG2 cells [134].

Lin et al. reported a bandgap engineering strategy to develop an intrinsically Raman-PA active probe that is based on semiconducting conjugated polymers agent polypyrrole (PPy) (Figure 8) [135]. This dual modal probe was prepared, in a one-pot reaction, by doping the semiconducting conjugated polymer with PDA. An enhancement of the Raman scattering and of the PA amplitude of PPy-PDA hybrid by 3.2 and 2.4 times. According to the authors, such a dual-enhancement effect in the hybrid was achieved by infusing these two macromolecules at the nanoscale, so decreasing the optical bandgap energy. The combined Raman-PA detection was applied both to in vitro and in vivo imaging, demonstrating that the understanding of the molecular structures and chain packing in hybrid semiconducting polymers-MNP can lead to the further development of novel materials with enhanced imaging abilities.

## 3. MNP Based Therapy and Theranostics

Therapeutic application of MNP take advantage of their unique properties. As mentioned MNP present a unique combination of feature such as biocompatibility, AOX action, broad spectrum light absorption and ultrafast excited state thermal deactivation that make them intrinsic therapeutic agent for AOX therapy and PTT. Additionally, thanks to the simplicity and versatility of covalent and non-covalent functionalization and loading, MNP can be tailored to achieve other function like the delivery of therapeutic agents such as drug molecules for chemotherapy or PS for PDT. Responsivity to different stimuli as local pH, ROS concentration and in particular light can be exploited to generate multiple activities. For example photo-thermal effect can be exploited for PTT but can trigger, simultaneously, drug release [136]. This multi-responsivity has been recently exploited to design MNP suitable for combined and synergistic therapy as well as for multi-modal imaging. In the next sections we will discuss most recent application of MNP to nanomedicine as therapeutics agent starting from simpler AOX therapy and drug delivery and moving gradually to more sophisticated systems up to theranostic nanoplatform for multi-modal imaging and synergistic multi-therapy.

### 3.1. Antioxidative Therapy

Reactive oxygen and nitrogen species (RONS) play important roles as signaling units in fundamental physiological processes [78]. Nevertheless, oxidative stress, hence the excessive formation of RONS, can contribute to the development of many diseases such as cancer, diabetes, autoimmune disorders, cardiovascular, and neurodegenerative diseases such as Parkinsonism [137]. Because of this duality, antioxidative therapies, aiming to balance RONS generation, are gaining increasing attention as possible strategies for preventing and treating oxidative stress-associated diseases. Nano-antioxidants possess several advantages in comparison to conventional molecular AOX, including higher antioxidative stability, stronger tolerance to harsh microenvironments and multiple free radicals targeting capacity [138].

Natural eumelanin acts as an endogenous AOX protecting the human organism from oxidative stress and, similarly, MNP possess excellent antioxidant activity and free radical scavenging capability. For this reason, their use in antioxidative therapy has been proposed to scavenge excessive RONS and supplement the endogenous ROS antioxidant defense mechanisms. Liu et al., for example, demonstrated the use of PEGylated-MNP in antioxidative treatment for the protection of the brain damage in ischemic stroke [34]. Knowing that oxidative stress and inflammatory activation, induced by the excessive production of RONS, are crucial for brain injury in ischemic stroke, the authors proved that the bio-inspired MPN can significantly decrease the infract area of the ischemic brain in rat model. These results proved not only the broad AOX activity of the MNP against various primary and secondary RONS but also their neuroprotective and anti-inflammatory activity. More specifically, MNP were demonstrated to be able to suppress the expression of inflammatory mediators and cytokines and alleviate RONS-triggered inflammatory responses. This first investigation inspired further development of the therapeutic uses of melanin in RONS-associated diseases. Bao and colleagues developed an efficient PDA-based nano-antioxidant for the suppression of ROS responsible for oxidative stress-induced periodontal disease [139]. Spectroscopic results revealed that the synthetized PDA NP exhibited broad and excellent antioxidative activities against toxic ROS, especially against hydroxyl radical and superoxide radical in vitro. In particular, experiments on human gingival epithelial cells proved that the nanoparticles were able to protect the cells from oxidative stress and inflammation reactions. Moreover, in vivo results, using a murine periodontitis model, clearly demonstrated that post-subgingival injection of PDA could efficiently remove ROS and decrease local periodontal inflammation without any side effects. Last but not least, the nanoparticles underwent biodegradation after ROS scavenge and their low systemic toxicity suggested their use not only for the improvement of local periodontal microenvironment, but also as an AOX for wide biomedical usages. In another example, Sun et al. developed ultrasmall Mn^2+^-chelated melanin nanoparticles incorporated in polyethylene glycol as a multi-scavenger, efficient AOX defense platform for treating murine acute kidney injury (AKI) (Figure 9) [140]. The kidney is very vulnerable to excessive oxidative stress and human embryonic kidney 293 cells were chosen as an in vitro model, providing proof for the broad and efficient ROS-scavenging properties of the nanoparticles as well as of their biocompatibility. Moving one step forward, in vivo, this effective and robust nano-antioxidant, after intravenous injection, displayed an excellent circulation and renal accumulation in AKI mice, alleviating significantly kidney damage. All things considered, the suggested nanoplatform can be qualified for AKI treatment via antioxidative protection, suggesting as well their potential use for other injuries associated with oxidative stress. As it is reported previously, MNPs possess also anti-inflammatory effect through the elimination of the ROS generated during inflammatory responses and the downregulation of pro-inflammatory cytokines.

Zhao et al. investigated the therapeutic capacity of PDA NP on acute lung injury (ALI) and peritonitis, pathologies that represent simple forms of acute inflammatory responses [141]. At cellular level, PDA NP were able to reduce significantly intercellular ROS in response to inflammatory stimulation. Meanwhile, in-vivo experiments revealed that a single dose of nanoparticles was sufficient to substantially relieve peritonitis and ALI in mice. In fact, the anti- inflammatory effect of the NP could be substantiated firstly by the reduced ROS generation and inflammation-related cytokines expression and secondly, by the decreased neutrophil infiltration and attenuated morphological tissue alterations, as in ALI they effectively ameliorated lung morphological changes. Undeniably, natural melanin, produced by living organisms, constitutes a potential and advantageous alternative to synthetic melanin. In particular, human-derived melanin eliminates the problem of biocompatibility as it is naturally present in the organism. Based on this, Hong and his colleagues extracted melanin from human hair for the construction of a multifunctional enzyme-mimicking nanoplatform [61]. The natural MNP were extracted by using alkaline degradation method and they were further cross-linked by coordination of metal ions like Fe, Cu and Mn. Different metal-chelated nanoparticles exhibited different AOX enzyme-like activity and different biomedical applicability. As regards antioxidative therapy, Cu-bound nanoparticles were found to serve a three enzyme-like activity (POD, SOD, CAT) simultaneously, mimicking a complete antioxidant chain that protects the cells against oxidative damage. Local induced inflammation in BALB/c mice was significantly reduced by the effective inhibition of the production of inflammatory cytokines and the admirable ROS scavenging activities of the NP, suggesting their therapeutic potential in oxidation-related diseases.

Although, as demonstrated, MNP play an intrinsic antioxidant action, their activity was in some case expanded by the integration of additional molecular AOX: Li et al., for example, proposed a novel combination of modified PDA NP with an exogenous AOX, resveratrol [142]. Resveratrol is a well-known natural AOX with strong free radical-scavenging activity, that unfortunately present the same disadvantages as other polyphenols: low solubility, unfavorable pharmacokinetics and biological instability. Its incorporation into the nano-delivery system was hence proposed in order to stabilize and enhance its AOX-activity. Zein NPs were firstly synthesized and loaded with resveratrol via phase separation method and then they were grafted with PDA and later with casein in order to increase the drug stability under different kinds of environmental stress (pH, salinity, UV irradiation). The resulting core−shell zein-PDA-casein nanocomposites exhibited good stability in physiological fluids as well as good biocompatibility and easy cellular uptake. Most importantly, the combined resveratrol-PDA assembly showed a five-fold enhancement of AOX activity and it was demonstrated to eliminate more exogenous ROS in comparison to free resveratrol. As a main advantage, this synergistic strategy could significantly reduce the administrated dosage of resveratrol, and it can be, at least in principle, extended to other polyphenols that exhibit cytotoxic effect at high dosage, providing new opportunities for defending oxidative stress.

### 3.2. MNP as Nanocarriers for Simple Drug-Delivery

Beside biocompatibility other features make MNP very promising as a novel class of drug delivery systems, particularly important are: (1) the possibility to manipulate the size and surface characteristics of the particles, (2) the tunable interaction melanin with the hosted molecules to achieve controlled or sustained release of the drug, (3) the AOX action of melanin that preserve the drug as from chemical degradation, (4) the target-specificity that can be obtained by attaching targeting ligands to the MNP surface, and, last but not least, (5) the possibility of administration of the drug-loaded MNP through various routes [6,8]. As far as the drug-NP interaction is concerned different strategies can be exploited to load MNPs such as, for example, covalent binding [143], nevertheless non-covalent drug encapsulation is surely the most convenient and versatile approach. Several drugs, in fact, possess strong binding capability to melanin, creating drug–melanin conjugates through the π-π interaction between the aromatic rings of the drug and segments on the surface of melanin as dihydroxy-indole and indolequinone [56]. Despite the fact that the drug-delivery properties of MNP have been exploited mostly in cancer treatment for chemotherapy, other examples of more common drugs have been reported and they will be discussed first.

#### 3.2.1. MNP for the Delivery of Non-Cancer Drugs

Some simple anti-inflammatory drugs (e.g., ibuprofen) have been used as model molecules to demonstrate the versatility of MNP as drug-delivery nano-agents. Li et al., for example, fabricated PDA coated TiO_2_ nanotube arrays (TNTs) via the ultrasonic method as drug-releasing materials for local drug delivery [144]. The presence of PDA in the PDA-modified TNTs was demonstrated to be crucial as it significantly enhanced the drug loading capacity and biomineralization ability of the TNTs, as well as it encouraged the nucleation of hydroxyapatite on the surface of samples. Indeed, when the nanotubes were loaded with ibuprofen (Ibu), a sustained drug release was achieved by adsorption or covalent bond interaction of PDA with the drug. In vivo results indicated an increase in the biosafety of the nanotubes after functionalization with PDA, as required in view of the application of these materials in bone implant therapies. Lim and coworkers developed a pH responsive type of MNP for combined small drug delivery with different kinetical profile as demonstrated by comparative release kinetic analysis [145]. Mesoporous silica nanoparticles were synthesizing via sol-gel method in presence of fluorescein isothiocyanate (FITC) and 3-aminopropyltrimethoxysilane (APTM), then they were coated with PDA and graphene oxide (GO) in order to increase loading capacity resulting from π-π interaction. NP were then loaded with two model drugs, i.e., ibuprophen (Ibu) and acetoaminophen (AAP). The system resulted to be an effective pH dependent drug delivery agent, in which release efficiency was controlled by drug-nanocarrier interactions. In particular, AAP release was rather pH independent thanks to the weaker interaction with nanoplatform, while Ibu transmission rate became significant only in acidic condition (pH 5.5). Yegappan et al. reported an example of incorporation of PDA NP in hydrogel for application in nanomedicine. In this case NP were loaded with dimethyloxalylglycine (DMOG), a proangiogenic small molecular drug, then thiol-functionalized hyaluronic acid (HA-Cys) was prepared by coupling cysteamine to hyaluronic acid after activation with 1-Ethyl-3-(3-Dimethylaminopropyl) Carbodiimide-N-hydroxysuccinimide (EDC-NHS) [146]. Finally, HA-Cys was successfully crosslinked with loaded PDA via Michael-type addition, forming a hydrogel that was reported to enhance endothelial cell migration, proliferation and attachment. Importantly, a sustained-DMOG release was achieved for a period of seven days, which promoted capillary tube-like structure formation of human umbilical vein endothelial cells (HUVECs), demonstrating the angiogenic potential of the drug. PDA has been also successfully employed in the treatment of the neurodegenerative disorders: Sardoiwala et al., proposed encapsulation of metformin (Met) in PDA for the treatment of Parkinson’s disease (Figure 10) [147]. Metformin behaves a potent neuroprotective agent but its application is limited because of the low bioavailability and potential risk of inducing lactic acidosis. These drawbacks were overcome by the encapsulation of the drug in PDA. In fact, the Met-encapsulated PDA nanoparticles demonstrated excellent neuroprotective potential and anti-Parkinsonism effect through various mechanisms that are namely: the upregulation of enhancer of zeste homolog 2 (EZH2) expression, proteasomal degradation of aggregated phospho-serine 129 (pSer129) α- Syn, suppression of apoptosis and anti-inflammatory activities as well as oxidative stress reduction. Additionally, PDA behaved as a dopamine replenisher as the loss of dopamine is strictly related to the pathology of the disease and as a free radical scavenger, further enhancing the efficacy of the treatment.

#### 3.2.2. MNP for the Delivery of Anti-Cancer Drugs

New-developed cancer treatment methods have achieved satisfactory results against various types of cancer, nevertheless, multidrug resistance is still a huge challenge in cancer therapy. Additionally, the effective and controlled delivery and release of drugs specifically at tumor sites is crucial for the achievement of the desired therapeutic outcome and to minimize systemic toxicity and side effects. Different therapeutic strategies based on MNP were recently developed to overcome the aforementioned obstacles, promising extraordinary therapeutic outcomes. PDA, according to Nieto et al. plays a primary role in and treatment of cancer, being per se an antineoplastic system, meaning it selectively kills cancer cells, especially in the case of breast cancer, without causing toxic effects to healthy cells [148]. A novel type of hollow PDA (HMPDA) was reported by Tao et al. who demonstrated that morphology control of PDA and their surface properties are fundamental for nanomedicine applications [149]. Drug adsorption capability is in fact strictly correlated to PDA surface area and their strategy produced PDA with enhanced surface area, granting an improved drug loading. The authors synthesized their HMPDA microcapsules via template method, using silica NP as template. PDA polymerization on silica NPs was driven by π–π interactions of trimethyl-benzene and polydopamine, and the final product was obtained after silica-core removal by 2% HF solution. HMPDA exhibited good adsorption properties, in particular for methylene blue and doxorubicin (DOX), an anticancer drug that can cause cancer cell death by inhibiting DNA topoisomerase II, and good sustained-release effect. Further studies on HMPDA was carried out by Tran et al. who showed that silica cores could be simply removed in water and, in particular, that the time needed for the removal is dependent on temperature and duration of the PDA coating process [150]. Similarly, Ozlu et al. developed polyethylene glycol (PEG) conjugated PDA for DOX delivery, demonstrating that PEG covalent surface modification led to an increased stability in blood circulation, and to a dramatic reduction on NP dimensions (15 ± 2.2 nm) in comparison with other studies [151]. DOX delivery resulted to be diffusion controlled.

In order to improve selectivity and achieve targeted delivery, a different strategy was reported by Zhao et al., who coated Paclitaxel (PTX) NPs with PDA and modified them with alendronate (ALN) as a ligand for osteosarcoma treatment [152]. PTX is a widely used anticancer drug since it has been proven to be an adjuvating agent for the immune system response. An in vitro assay demonstrated enhanced cytotoxicity against osteosarcoma cells and in vivo distribution analysis indicated enhanced delivery properties. As it was mentioned before multidrug resistance (MDR) is one of the biggest challenges in cancer therapy that is difficult to tackle also because of the high heterogeneity of this pathology. Mitochondria, which are strictly related to the process of carcinogenesis acting as a cell powerhouse, were chosen as a target by Li et al. in order to overcome MDR [42]. More in detail these authors constructed mitochondria-Targeting PEGylated PDA nanoparticles for the delivery of DOX by using triphenylphosphonium (TPP) as the mitochondrial penetration moiety. Repeated treatment experiments on MDA- MD-231 cancer cells showed that the formed DOX-loaded PDA-PEG-TPP NP possessed elevated toxicity and higher potential to reduce drug resistance compared to simple DOX-loaded PDA-PEG nanoparticles. It is worth noticing that in this multicomponent system PDA acted not only as an excellent transport agent, delivering effectively DOX into mitochondria, but also actively damaged mitochondrial respiration by causing modifications on the subunits of complex I and complex III of the electron transport chain. More recently, a novel approach was reported by Singh et al. for the delivery of the same anti-cancer drug DOX [153]. In this case, PDA was exploited for coating the surface of iron oxide magnetic NP designed to bind taurine, a free sulfur-containing ß amino acid (Figure 11). Taurine conjugation improved the biocompatibility and delivery efficacy of the nano-carrier as it has the potential to cross the blood brain barrier. Moreover, thanks to the interaction between the sulfonic head group of taurine and the protonated amine of DOX, this drug molecule was bound to the PDA-coated iron oxide nanorods very efficiently, yielding to a loading capacity as high as 70.1%. Moreover in vitro experiments on human prostate adenocarcinoma PC-3 cells revealed good cellular uptake and pH-dependent drug release able to successfully ablate cancer cells. Although MPN have been suggested by some authors for the bare delivery of anti-cancer drugs, more sophisticated multi-therapy approaches that combine drug delivery with other therapeutic techniques, e.g., PTT, have been recently proposed in order to ablate tumors more efficiently and reduce the possibility of cancer re-occurrence, as will be discussed in next sections.

### 3.3. Photothermal Therapy (PTT)

Photothermal therapy (PTT) has earned much attention as a minimally-invasive therapeutic strategy for cancer treatment and it is based on the local conversion of photon energy into heat, sufficient to destroy cancer cells [154]. More specifically, this therapy exploits photothermal transduction agents (PTA) that absorb the irradiated light and convert it into heat increasing the temperature of the surrounding environment [155]. Much of the uniqueness of PTT is due to its spatio-temporally controlled photothermal action. In fact, the specific therapeutic effect is rather independent of the cell type and specificity is achieved by limiting the light-triggered region, with the advantage of causing no systemic toxicity. Additionally, specific targeting allows to minimize the damaging the surrounding non-targeted tissues. PTT can be, to a large extent, considered as a branch of nanomedicine, since most of PTA are nano-scaled [156], and within these MNP play an important role since they fulfill the main requirements of an ideal PTA. In fact MNP, beside the high biocompatibility, show high photothermal conversion efficiency, ultrafast excited-state thermal deactivation, large absorption coefficient in the near infrared region (NIR) and the possibility for easy surface modification [157]. NIR region is of significant biological importance and provides a region in the spectrum useful for therapeutic applications as the tissue components show minimum absorption in this range of wavelengths, enhancing the tissue penetration, the targeting efficiency as well as the therapeutic effect [158]. In vivo and in vitro applicability of MNP to PTT have been demonstrated to a large extent, in the past, and recent efforts have been mostly dedicated to develop system with enhanced performances [159,160]. Yang et al., for example, investigated the feasibility of arginine-doped synthetic MNP for PTT [161] Compared to conventional PDA NPs these novel arginine-containing nanostructures showed improved total photothermal effect. According to the authors, this resulted from: (i) the enhancement of the absorption of NIR light due to the formation of donor−acceptor microstructures and (ii) to a further deactivation of the already poorly efficient non-thermal radiative transition processes as a consequence the increase of free radical concentrations. Interestingly, even a minimal doping with arginine could increase the total photothermal efficiency up to almost 60%, an improvement was verified not only in vitro but also in vivo. In fact, intravenous administration of the nanoparticles in 4T1 tumor bearing mice led to significant tumor temperature elevation and additional tumor growth inhibition upon NIR irradiation A different approach took advantage of the excellent metal chelation ability of melanin, for the production of metal-chelated MNP for PTT imaging-guided therapy. This strategy was proposed by two different research groups which followed two different synthetic schemes for the construction of Mn-chelated melanin NPs. In particular, in the former example by Liu et al. followed a one-pot intrapolymerization strategy to fabricate manganese-eumelanin coordination nanocomposites through the chemical oxidation-polymerization of 3,4-dihydroxy-DL-phenylalanine (DL-DOPA) precursor with KMnO_4_ as a manganese source [94]. Due to their excellent biocompatibility, strong NIR absorption and high Mn-chelation ability, the occurring NPs were found to be extremely efficient for tumor PTT guided by dual-modal imaging (MRI and PAI). An additional advantages of this one-pot preparation method was its simplicity and convenience, especially in comparison to the multi-step process reported in the latter example by Sun et al. [162]. These authors, in fact, proposed a post-polymerization doping strategy, where the transition metal ions were chelated in a second step onto the first-synthesized MNP. Despite this preparative complication it was substantiated that the tailor-made NP could be successfully employed for combined imaging-guided PTT. In similar fashion, PDA was used as a bio-template for the synthesis of a copper sulfide nano-PTA followed by further chelation of iron ions [163]. The resulting so-called CuPDF NP were reported to be ideal for MRI guided photothermal cancer therapy as they showed an enhanced photothermal efficacy, compared to pure PDA nanoparticles, together with an evident MRI signal enrichment. In vivo, these metal doped MNP exhibited superior biocompatibility and produced rapid temperature increase, upon NIR laser irradiation, adequate for the ablation of tumors. As shown, one of the advantage of the incorporation of metal ions into MNP was the achievement of multifunctional agents that combine multiple diagnostics with PTT activity. Indeed, this multi-functionality was further expanded, and hybrid melanin-metal NP with additional properties were developed. For example, given the strong affinity of melanin to metal ions, gallium-based liquid metal (LMs) nanodroplets were fabricated by using MNP as capping agents [164]. In this case, a shape-controlled synthesis of different LMs nanostructures was achieved by simply modulating some reaction parameters such as the concentration of the melanin, the sonication period and the temperature. From all the emerged nanostructures (namely nanorices, nanospheres, and nanorods), eutectic gallium indium (EGaIn) nanorices presented the best photothermal conversion efficiency in conjunctions with magnificent photostability and biocompatibility. This conclusion was based not only on in vitro, but also on in vivo trials carried out in Balb/c mice, that demonstrate the efficacy of (EGaIn) nanorices as phototherapeutic agents for PTT Investigating the relationship between PTT and apoptosis in prostate cancer cells, Kong et al. took advantage of the properties of PDA-coated branched Au–Ag nanoparticles (Au–Ag@PDA NPs) [165]. Photothermal action was found to be able to cause oxidative stress in cancer cells and to activate the mitochondrial-related apoptosis pathway, rather than cause necrosis, by increasing the expression of the most important pro-apoptotic protein, BAX. Clear differences in PTT effects were observed in treated androgen-dependent (LNCaP) and androgen-independent (DU145) cells, being the first more sensitive to the Au–Ag@PDA NPs plus laser treatment that produced higher death rate. Furthermore, pre-administration of BAY 11-7082, an inhibitor of NF-κB signaling pathway closely related to tumor cell resistance, promoted apoptosis in the hormone-independent DU145 cells, increasing their susceptibility to the heat treatment, both in vivo and in vitro. Although metal-containing NP described above in this section are based on synthetic melanin, natural melanin NP constitute a valuable class of melanin-based photothermal agents. Remarkably, natural melanin, extracted from living organisms, has attracted much attention due to its native biocompatibility and biodegradability that contribute to the mitigation of side effects in humans. As mentioned, sepiomelanin, isolated from the ink sac of cuttlefish, is considered to be an easily accessible analogous of eumelanin of mammals. Moreover it exhibits a very rapid and strong photothermal response upon NIR-irradiation [166], and an enhanced photothermal efficiency with respect to artificial melanin-like PDA NPs [167]. In the light of this, Wang et al. took advantage of melanin extracted from ink of sacrificed cuttlefish for the construction of Au-decorated melanin (Au-M) nanocomposites [168]. These Au-M NP, arising from the in-situ growth of Au nanoparticles on natural melanin, possessed high photothermal conversion efficiency and biocompatibility. Taking into account that the extracted melanin has high biocompatibility, due to its completely biosynthetic nature, the authors further demonstrated that the Au decoration did not mitigate this exceptional benefit. Additionally, in tumor bearing mice, post-injection signals of the tumor centers clearly demonstrated the efficiency of the NP as contrast agents for computed tomography (CT) and PA imaging. Thermal cell necrosis accompanied with cancer cell shrinkage, loss of contact, eosinophilic cytoplasm, and nuclear damage provided more insight of the utility of the natural derived NPs for multi-imaging guided PTT. In a like manner, Kim et al. evaluated the applicability of natural melanin-loaded nanovesicles (melasicles) decorated with phospholipids as nano-PTAs for NIR mediated cancer therapy [169]. The amphiphilic phospholipids, being the main component of the cell membrane, conferred to the nanovesicles membrane-like characteristics contributing to the reduction of immune response. The sol-gel behavior of the melasicles suspension permitted the application of this polymeric preparation in the form of injectable hydrogel into cancer regions. After intravenous injection into CT26-bearing mice through tail veins and upon NIR irradiation, melasicles demonstrated to emit sufficient heat to provoke immediate death of cancer cells through photo-heat conversion effects. Furthermore, post treatment analysis, after a period of two weeks, demonstrated significant tumor suppression, or even total elimination, especially upon double laser irradiation, indicating the superior properties of the nanovesicle for PTT.

Aiming to further mitigate the already low cytotoxicity of melanin, Zhang et al. reported the synthesis of a multifunctional melanin-based nanoliposome (Lip-Mel) for imaging-guided photothermal ablation of tumors [170]. In particular, according to these authors, the unique core/shell structure was made by components originating from human body, having liposome-PEG as the shell and melanin granule as the core. Moreover, encapsulation nurtured the biosafety of melanin further decreasing its cytotoxicity in comparison to the pure biopolymer and enhanced the photothermal performance, the biological stability, and the intensity of the diagnostic signals. The encapsulated melanin NP could efficiently accumulate into the tumor region via enhanced permeability and retention (EPR) effect, ablate photothermally the cancer tissue and act also as contrast agents for concurrent PAI and MRI. In order to achieve targeted cellular uptake of MNP Zhou and his group focused on autophagy, (Figure 12) a process that represents an intracellular catabolic pathway responsible for the degradation and recycling of biomacromolecules and damaged organelles thought to be close related to cancer [171]. Beclin 1, a mammalian homologue of yeast Atg6/Vps30, initiates autophagy pathway acting also as tumor suppressor. Taking this into consideration, Beclin 1-conjugated MNP were fabricated in order to promote the autophagy activity in cancer cells and further assist tumor growth regression capacity of PTT. More in detail, the NP were composed by PDA NP conjugated with beclin 1- derived peptide, polyethylene glycol and cyclic Arg-Gly-Asp (RGD) peptides. The role of RGD was fundamental for the selectivity of the therapy, as it favorably bound to cancer cells overexpressing integrin α_v_β_3_, resulting in enhanced internalization and accumulation of the NP in tumor sites. An in vivo study on a mice model bearing MDA-MB-231 tumors revealed that the autophagy up-regulation associated to PTT exhibited high efficacy in the regression of tumor growth at mild treatment temperature around 43 °C.

As mentioned before, systemic blood circulation, tumor accumulation as well as the ability to cross the biological barriers are of main importance for effective PTT. External decoration can enable the NP with improved blood circulation time; minimizing immune recognition. In this framework, erythrocyte (RBC) membrane coating represents a biomimetic approach to prolong blood circulation of NP due to the presence of “maker-of-self” protein, CD47, on their surface responsible for the release of “don’t eat me “signals. In a recent study, Jiang et al. fabricated erythrocyte-cancer hybrid membrane-camouflaged MNP (Melanin@RBC-M) by the fusion of RBC membrane with MCF-7 cell (human breast cancer cell line) membrane for in vivo PTT [172]. Both membrane coatings played a crucial role in the operation of the photothermal agent. In more detail, this RBC-M hybrid membrane vesicles endowed melanin with long circulation, due to RBC membrane, and homotypic tumor targeting, due to MCF-7 cell membrane. Different protein weight ratios of the two membrane components have been investigated for optimizing the membrane-camouflaged nano-PTA. Considering the inherited photothermal property and biocompatibility of melanin core, the NP with membrane protein weight ratio 1:1 manifested the best and superior PTT efficacy, due to the optimal balance between prolonged blood circulation and homotypic targeting. PTT finds application also different from cancer therapy, and it was also adopted as a non-invasive ablation therapy for chronic brain diseases. According to Liu et al., PDA NPs can be harnessed as promising candidate for nanoparticle-assisted PTT against temporal lobe epilepsy and other nontumorous central neurological diseases [173]. To sum up, the straightforward approaches illustrated the significance of MNP as photothermal agents for PTT, providing personalized treatment alternatives with high therapeutic efficacy and multifunctionality.

### 3.4. Synergistic PTT

The heterogeneity and complexity of many diseases, including cancer, has raised the need for the development of innovative therapeutic approaches. In particular, the unsatisfactory results of different monotherapies led scientist to move towards new synergistic therapeutic approaches with promising clinical outcomes and performance [174]. As regards cancer, in many cases, single treatments are unable to eliminate the whole tumor and to prevent cancer metastasis. Indeed, recent advances in cancer therapy have gradually shifted from a focus on monotherapy to combined therapy. Combined therapy is based on the cooperative enhancement interactions between two or more therapeutic treatments that are able to generate superadditive therapeutic effects, known as “1 + 1 > 2”, meaning that the combined therapeutic effects are much stronger than expected considering the theoretical combination of the corresponding individual treatments. In the view of this, advanced nanotechnology focuses on the design of multifunctional nanomaterials for the co-delivery/co-assembly of two or more therapeutic agents for the integration of multiple therapeutic modalities within a single nanoplatform [175]. The exceptional and numerous properties of MNP suggested the development of several multifunctional nano-vectors for combined and synergistic therapies as detailed below.

#### 3.4.1. PTT-Immunotherapy

Immune escape is recognized as one of the key hallmarks of cancer. This finding inspired scientist to move towards strategies that are able to restore immunity for cancer prevention and cure. Immunotherapy is an indirect cancer treatment which is based on the power of the host immune system and emerging technologies associated with immunotherapy comprise a very promising alternative for cancer therapy [176]. Considering that PTT can cause immunogenic cell death, new MNP-based nanoformulations are engineered for immunotherapy-PTT combined therapy. More in detail, photothermal ablation of tumor using multifunctional NP, that act like PTA and immune-response activators or immune-adjuvants, is capable to generate strong anti-tumor immunological effects for effective cancer immunotherapy. This synergistic therapy serves several advantages as it is not only able to eliminate primary tumors, but also to attack and kill spreading metastatic tumors and prevent their reoccurrence by offering an immune-memory protection [177]. Dopamine is one of the key neurotransmitters as well as an important connector between the nervous and immune system. As an extracellular messenger, dopamine regulates the immune system by interacting with dopamine receptors on the immune cells. Due to the high distribution of dopamine receptors in tumors, dopamine exhibits antiangiogenic and anticancer activity via activation of dopamine receptor on endothelial and tumor cells. Being a self-polymer of dopamine, PDA was reported to serve an excellent candidate for antigen delivery carrier for tumor single immunotherapy [178]. Antigenic peptides are widely utilized in immunotherapy because of their advantageous properties, such as: (i) direct functional T cell epitopes, (ii) low toxicity, (iii) low cost, and (iv) ease of synthesis. In particular, the immunotherapy mechanism is based on synthetic peptides presentation. In order to get this fatal prerequisite, antigen presenting cells (APC) have to capture a proper amount of the desired antigen, which has to remain in cytoplasm for the time needed. At this point, antigens, like antigen-ovalbumin, promote cross-presentation within APCs activating MHC I mediated presentation, and finally induce cellular immunity. Considered the previous results that demonstrate the applicability of MNP for immunotherapy and PTT separately, Chen et al. investigated the MPN efficiency in combined therapy by integrating the two treatments in a single platform. The suggested NP, named pD-Al_2_O_3_, were made by an inner core of Al_2_O_3_ NPs and an outer shell of PDA [179]. The pD-Al_2_O_3_ NP were injected directly into tumors of mice with B16F10 allografts. Due to their high photothermal efficiency, under NIR laser irradiation of the animals, the majority of tumor tissues were killed by PTT, resulting also in the release of tumor-associated antigens. The triggered systemic immune response aimed in the elimination of residual tumor cells. Indeed, the Al_2_O_3_ contained within the NP, together with a co-administered widely used, inexpensive adjuvant, CpG, effectively triggered robust cell-mediated immune responses that could help eliminate the residual tumor cells and reduce the risk of tumor recurrence. In another case, Rong et al. developed an iron chelated MNP-based therapeutic agent to counteract the immunosuppressive tumor microenvironment [180]. Tumor-associated macrophages (TAMs), which are actively recruited in tumors, are one of the main drivers of this immunosuppression.

More in detail, TAMs predominately exhibit an immunosuppressive M2-type function, promoting cancer initiation and malignant progression. For the aforementioned reason, these authors developed a method aimed in the repolarization of M2 macrophages into M1, which on the contrary are potent effector cells that could directly kill tumor cells, elicit antitumor immunity and antagonize the immunosuppressive activities of M2 macrophages. Importantly, the iron chelated PEGylated PDA NP could robustly repolarize M2-like TAMs towards M1 mode, which acted as primary phagocyte and antigen presenting cells in tumors, both in vitro and in vivo. Possessing high biocompatibility and photothermal conversion efficiency, the novel MNP were able to hyper-thermally kill cancer cell, via PTT and additionally provoke the release of tumor associated antigens (TAAs). These TAAs released by the hyperthermic cell death could be captured, processed and presented by the repolarized M1 macrophages through the major histocompatibility complex class II (MHC II) pathway, recruiting T-helper cells and effector T cells in tumor site. As result, the synergistic therapy could effectively reverse the immunosuppressive tumor microenvironment and significantly control tumor growth in multiple tumor models in mice, extend survival rate and further protect them against malignant metastasis in late stage breast cancer. A similar approach, but based on the use of natural melanin-containing cuttlefish ink nanoparticles (CINPs), was followed by Deng et al. who demonstrated an efficient synergistic PTT-immunotherapeutic approach for tumor growth inhibition [181]. In particular, CINPs reach in melanin, containing also various amino acids and monosaccharides, were extracted from the ink sac of cuttlefish using a differential centrifugation method (Figure 13). Having the same therapeutic action of the previously-cited iron chelated PDA nanoparticles, CINPs exhibited excellent ability to repolarize M2 TAMs towards antitumor M1 phenotype and further promoted the infiltration of cytotoxic T cells within the tumor in vivo. Similarly, due to the outstanding photothermal properties and strong photostability of the nanoparticles, PTT-induced tumor killing was actualized with additional release of TAAs in situ. In short, the combination of PTT with tumor-immunotherapy almost completely inhibit tumor growth accompanied with enhanced immune responses and elevated production of various antitumor factors (TNF-α, INF-γ, IL-6, IL- 12, NO), leading not only in the suppression of primary tumors growth, but also in the hindrance of lung metastasis in CT26 tumor bearing mice. Cuttlefish extracted MNP, were also chosen by Li et al., who investigated the applicability of multifunctional membrane-camouflaged nano-therapeutic agents [182]. The extracted MNP were coated with 4T1 cancer cell membrane to achieve homologous adhesion of tumors and escape immune clearance. Upon reaching the tumor site, the “smart agents” acted as superior PTA resulting in PTT-induced immunogenic cell death (ICD). In fact, ICD provoked the release of damage-associated molecular patterns (DAMPs), that could provide ‘‘eat me’’ signals for the innate immune system to increase the immunogenicity. Additionally, the intra-peritoneal injection of an immunoblocking inhibitor, IDOi, exhibited a synergistic effect with PTT-induced ICD, inhibiting the tumor immune escape and enhancing the antitumor immune response. This combined treatment resulted in an increased amount of CD^8+^ T cells and a higher level of cytokines, which eventually led to a strong and persistent antitumor immune response for an enhanced therapeutic effect against primary and abscopal tumors, in vivo. In a different application, a transdermal polymeric microneedle vaccine patch with target the antigen-presenting cells, was suggested by Ye et al. as a novel melanin-mediated immunotherapy strategy [183]. The proposed vaccine was composed by inactive B16F10 whole tumor lysates containing melanin, which were encapsulated into polymeric microneedles allowing the gradual release of the lysates upon insertion into the skin. The presence of melanin in the patch contributed in the local skin temperature increase, via remotely controllable NIR irradiation, causing the release of inflammatory cytokines and subsequently effectively recruiting and activating immune cells at the vaccination site, through the uptake and presentation of TAAs by dendritic cells. In more detail, PTT- vaccination synergy, increased the infiltration of polarized T cells and the local cytokine release. In total, the results suggested the potent anti-tumor efficiency of the photo-combined responsive vaccine toward established primary and distant tumor and further established its preventive role towards tumor engraftment in prophylactic models.

#### 3.4.2. PTT-Chemotherapy

Traditional chemotherapy (CT) exhibits several disadvantages, with the main being the generation of severe adverse effects in healthy tissues and the development of intrinsic multi-drug resistance by tumor cells. Nanotechnology, represents a vital revolution in oncology, providing alternatives to conventional chemotherapeutic drugs and suggestions for advanced multifunctional delivery systems [10,184]. The combination of chemotherapy with PTT has been reported to present as one of the most efficient synergistic strategies, as some anticancer drugs exhibit synergistic interaction with the PTT-induced heat or enhanced cytotoxicity upon high temperature. Consequently, the multifunctional nano-PTA can not only aid the drug delivery, but they can also improve the cancer cell-killing efficacy of chemotherapy and reduce the side effects [175]. PTT-enhanced chemotherapy is based on different approaches, all engaging a photothermal conversion nanosystem that also works as nano-carrier for drug delivery [174]. Most recent research is focused on the development of nanocarriers which can effectively deliver the therapeutic agents on the target site, without early leakage, and consequently release upon a specific stimuli, such as NIR irradiation also in combination with pH [41]. This behavior can be achieved by simple or modified MNP [151,185]. Combining drug binding capacity of MNP with their innate PTT efficacy, Zhang et al. developed an outstanding synergistic PTT-CT cancer treatment [186]. In particular 4-arm PEG amine modified PDA NP were utilized as multifunctional platforms for the delivery of the anticancer drug Cisplatin. In practice, the developed MNP were conjugated with the cisplatin prodrug Pt (IV)-COOH and they were found to exhibit a chemical reduction-responsive release of cytotoxic Pt (II). After the internalization of Pt (IV)-melanin like nanoparticles (Pt (IV)-MeNPs) by cancer cells, the intracellular tumor reductive or acidic environment could provoke the release of the anticancer drug, which was found to be higher upon NIR irradiation. The increased drug release by NIR irradiation combined with the high photothermal killing efficiency of Pt (IV)-MeNP, confirmed the synergistic effective PTT-CT antitumor activity both in vitro and in vivo. Likewise, melanin dots were used as active nanoplatforms able to afford the simultaneous loading of different modules, including Pt (II) metallacycle and fluorescent NIR-II molecular dye, for NIR-II/PA dual-modal imaging-guided synergistic CT-PTT [187]. The role of melanin dot platform in the developed molecular-dye-modified theranostic agent was crucial, as due to its preferential passive tumor accumulation, Pt (II) metallacycle was successfully delivered to tumor sites via an enhanced permeability and retention effect. Additionally, the nano-agents possessing high stability, good optical properties, passive targeting ability and two imaging modalities with superior signal-to-background ratios, could improve cancer diagnosis and treatment by complementary providing information in vivo. The combination of the antitumor activity of Pt(II) metallacycle with the inner photothermal properties of MNP enabled the synergistic therapy to decrease the desired antitumor effects, and, on the other hand, the side effects in U87MG tumor-bearing nude mice models.

Integrating PT-CT platform with imaging agents for imaging-guided therapy, is highly beneficial for precise treatment guidance and assistance. Between the different imaging modality Zhang et al. chose optical guidance achieved by replacing conventional platinum-based anticancer drugs with photo-luminescent iridium complexes [115]. These authors reported the preparation and application of an integrin-targeted MNP loaded with the phosphorescent anticancer iridium (III) complex LysoIr. The high tumor accumulation and high selectivity of the LysoIr@PDA-CD-RGD NP for integrin-positive human brain glioma U87 cells induced caspase-mediated apoptotic cell death through lysosomal damage and ROS generation upon NIR light irradiation. The release of the anticancer drug was demonstrated to be not only pH-responsive, but also photothermal heating-responsive. As an alternative to optical imaging, MRI was also proposed to expand the features of PTT-CT MNP-based systems by Wang et al. who fabricated a PDA based bone-targeting therapeutic nanoplatform with high selectivity recognized malignant bone tumor and osteolysis [188]. In this design, PDA NP were conjugated to alendronate (ALN), which significantly increased the affinity of the NP to hydroxyapatite and further enhanced their accumulation at the osteolytic lesions in comparison to pure PDA. When ferric ion (Fe)-doped PDA-ALN were synthetized, the presence of ferric ions revealed the effective tumor accumulation of the MNP. This was demonstrated by the magnetic MRI contrast of the bone tumor that was intensified because of the activity of the metal-chelated NP as T1 contrast agent. Moreover, dual stimuli-responsive release of the chemodrug 7-ethyl-10-hydro- xycamptothecin (SN38) was observed from the PDA-ALN nanocarrier, in response to pH change and NIR irradiation. The combined CT-PTT therapy efficiently suppressed the growth of bone tumors and diminished the osteolytic damage with better results than the individual treatments. By the same token, PDA-containing injectable hydrogel was employed for the cure of bone cancer, simultaneously promoting also the repair of bone defects [189]. The hydrogel was composed by PDA-decorated nano-hydroxyapatite (n-HA) NP loaded with cisplatin, which were introduced via Schiff base reaction between the aldehyde groups on oxidized sodium alginate (OSA) and amino groups on chitosan (CS). All the components played a key role in the system as n-HA presents a major inorganic component of bone tissues and OSA and CS possess high biocompatibility, biodegradability and similarity to bone matrix components. Synergistic anti-tumor effect was achieved through NIR irradiation, which initially caused local hyperthermia inducing tumor cells apoptosis and subsequently triggered cisplatin release due to the high temperature induced-breakage of hydrogen bonding. Furthermore, the injectable hydrogel not only suppressed the growth of solid tumors in vivo, but also aided the adhesion and proliferation of bone mesenchymal stem cells in vitro and stimulated bone regeneration in vivo in BALB/c mice. Recently Wang et al. proposed another approach for the construction of a thermoresponsive self-healing injectable hydrogen for PPT-CT synergistic therapy [190]. The preparation of the “smart” hydrogel engaged the formation of dynamic covalent enamine bonds between the amino groups in polyetherimide and the acetoacetate groups in the four-armed star-shaped poly(2-(dimethylamino)ethyl methacrylate-co-2-hydroxyethyl methacrylate) modified with tert-butyl acetoacetate and the further dispersion of PDA and loading of doxorubicin (DOX). Notably, after intratumoral injection the hydrogel exhibited thermoresponsive phase change and volume shrinkage upon NIR irradiation, a process that initiated the controlled release of DOX, leading to highly efficient thermo-chemotherapeutic anti-tumor effect in breast cancer model. In alternative to hydrogels, drug-loaded nanofibers were proposed as systems for local controlled molecular delivery by Obiweluozor et al. [191]. In this case, remote-controlled drug release and therefore cooperative PTT-CT was achieved by a single polydioxanone (PDO) nanofiber containing PDA and bortezomib (BTZ). The nanofiber, which was fabricated with electrospinning method, was expected to maintain direct contact with the tumor for continuous localized heat production and drug release. Most importantly, BTZ chemotherapeutic drug exhibited NIR and pH responsive release and enhanced cytotoxicity with PTT-induced hyperthermia. In combination to the effective cancer cell-binding capacity of the PDO nanofiber, this process resulted in efficient CT 26 colon cancer cells ablation. In another example, Yang et al. achieved to synthetize a new type of MoS_2_-based nanoplatforms with enhanced photothermal properties, excellent biocompatibility, good drug loading capacity and dual stimuli-responsive drug release [192]. In more depth, poly- ethyleneimine and PEG were utilized to modify the MoS_2_ nanosheets in order to afford a hyaluronic acid functionalization. Following DOX chemotherapeutic drug and melanin were loaded to the nanocarrier. The role of the biodegradable hyaluronic acid was crucial as it enhanced the targeting efficiency in high CD44 receptor expressing MCF-7 cells and facilitated the endocytosis of the nanocomplex by the cells. The combined loading of the chemotherapeutic and the photothermal agent in the nanocarriers enabled a drastic synergistic anti-cancer effect in nude mice bearing MCF-7 tumors compared to single treatments.

Encouraged by the excellent drug loading ability of MNP accompanied with their excellent photothermal conversion efficacy more and more researches investigated novel approaches to exploit at its best the synergy of PTT-CT combination. To improve the selectivity and targeting of the treatment, Li et al. reported a glucose transporter 1 (GLUT1)-targeting and simultaneously pH/NIR -responsive cytosolic drug delivery nanoplatform [193]. GLUT1 was exploited as a confirmed target for drug delivery which is over-expressed in fast-growing and metastatic tumors. In the view of this, glucosamine and amino diethylene glycol glucose, which were conjugated with PDA and further loaded with bortezomib (BTZ), were utilized as GLUT1 targeting ligands promoting selective accumulation and efficient entry of the anticancer drug in the tumor cells through GLUT1-mediated endocytosis. The endo/lysosomal pH and NIR irradiation promoted the robust release of BTZ resulting in excellent tumor destruction ability even with only one treatment. An alternative anticancer strategy that was found to exhibit excellent therapeutic outcomes in combination with PTT exploited the inhibition of Wnt/β-catenin signaling cascade [194]. In this case, melanin coated magnetic NP were loaded with a Wnt signaling inhibitor, obatoclax (OBX), through π-π stacking and hydrophobic interaction (Figure 14). The resulting MNP generated hyperthermia upon laser irradiation, which triggered the release of OBX drug and also dramatically enhanced the accumulation and internalization of the drug in tumor cells. Possessing MR/PA dual-modal imaging capability, these MNP made possible a multimodality imaging guided mild hyperthermia- enhanced chemotherapy, as NIR irradiation significantly increased OBX-mediated inhibition of the Wnt/β-catenin signaling. The anti-tumor efficacy of the treatment was further verified by the suppression of mouse mammary tumor virus (MMTV)-Wnt1 transgenic tumor via Wnt signaling pathway blocking.

In another study, a platelet-camouflaged nanodrug was designed for resistant cells and the tumor vasculature dual-targeting strategy [195]. MNP and DOX were encapsulated inside RGD peptide (c(RGDyC))-modified nanoscale platelet vesicles, which exhibited significant immune-escape capability and effective αvβ3 integrin-targeting. After intravenous administration and upon NIR irradiation the complex nano-cocktail inhibited the growth and metastasis of drug-resistant breast cancer through a chemo-photothermal mechanism, which derived from the biomimetic properties and the multi-target ability of the nanodrug. Following this, in a recent therapeutic approach by Chen et al., a pH- and photothermal-responsive zeolitic imidazolate framework (ZIF-8) compound was utilized as a nanocarrier for the simultaneous delivery of another green-tea deriving anticancer drug except for DOX, namely (–)-epigallocatechin-3-gallate (EGCG) [196]. The NP were constructed via the self- assembly of EGCG@ZIF-8 and self-polymerization of dopamine and were afterwards functionalized with PEG to achieve DOX loading via π-π stacking accompanied with improved biocompatibility. According to the results, synergistic PTT-CT was realized as both drugs were effectively released in the tumor microenvironment, provoking autophagic flux and hastening the formation of autophagosomes. Moreover, the tumor inhibition capacity of the combined therapy was significantly enhanced in comparison to treatment without irradiation, ablating remarkably in vivo tumors. Liu et al. developed dynamically PEGylate and Borate-coordination-polymer-coated PDA NP for DOX delivery and therefore synergistic therapeutic effect [197]. The coordination polymers (CP) owning a porous structure were synthetized using Zn (II) metal ion, boronobenzene-1,3-dicarboxylic acid (BBDC), and 1,3,5-benzene tricarboxylic acid (BTC). As a result, phenylboronic acid-functionalized CP layer was assembled on the PDA NP which were afterwards functionalized with PEG. Most importantly the presence of phenylboronic acid groups in BBDC caused “active tumor targeting” under weakly acidic condition acting as a ligand for sialic acid, which is highly overexpressed in tumor cells. On the other hand, PEG surface modification was responsible for “passive targeting”, which together with the prior one resulted in synergetic tumor targeting property and outstanding anti-cancer results both in vivo and in vitro. Since 2017, Xing et al. showed that PDA could be efficiently used in PTT-CT in order to fight multiple drugs resistance (MDR) cancer cells [198]. More recently, Guan et al. developed a potent anticancer nanodrug by using mesoporous PDA NP for delivery of sorafenib (SRF, clinically-approved drug) and superparamagnetic iron oxide (SPIO) NP in order to induce ferroptosis of tumor cells [125]. Ferroptosis is an iron-dependent type of programmed cell death characterized by the accumulation of peroxides. The formed nanodrugs showed excellent biocompatibility, high loading capacity and drug-release efficacy in response to numerous stimuli as pH, temperature and glutathione. It is worth noticing that ferroptosis alone exhibited limited anticancer effect. On the other hand, its combination with laser irradiation-triggered photothermal therapy effectively inhibited tumor growth, suggesting the synergic effect of the treatment.

Nanoliposomes have emerged as novel nanocarriers, and several smart responsive systems were produced sensitive to pH, light, enzymes, magnetism and heat. In this context, Wang aet al. demonstrated the utility of a melanin-containing nanoliposome for targeting a specific type of cancer, pancreatic cancer [199]. In their study, dual-functional, melanin-based nanoliposomes-gemcitabine (GEM) were synthesized, in order to combine chemotherapy and PTT for pancreatic cancer treatment. GEM is a novel chemotherapeutic drug for this kind of cancer, but it shows a lack of effective accumulation in pancreatic tissues, hence, thermosensitive nanoliposomes were prepared in order to exploit hyperthermia-triggered mechanism, via PDA encapsulation. Under NIR irradiation GEM release from nanoliposomes was controlled and enhanced via the PDA-induced hyperthermia.

#### 3.4.3. PTT-Gene Therapy

Gene Therapy (GT) is based on the use of therapeutic genes to treat cancer or other diseases. More in detail, it involves the therapeutic delivery of genetic material into patients’ cells in order to compensate for abnormal genes and to express specific proteins. So far, GT has been reported to cooperatively improve the effects of PTT resulting in outstanding synergistic anticancer efficacy [175]. In particular, short interfering RNA (siRNA), which are double-stranded RNA molecules able to specifically silence the expression of different genes, represent a breakthrough in GT and have been effectively applied lately in myriad of disease treatments [200]. Heat-shock-proteins (Hsps) are highly expressed in cancer cells conferring thermoresistance and impairing the hyperthermia-induced cell death, something that compromises the thermal ablation efficacy of PTT. Certain types of siRNA can silence the expression of Hsp proteins, like Hsp70, inhibiting in this way the heat shock resistance of cancer cells and subsequently increasing their susceptibility to PTT. In the light of this, Ding et al. proposed the encapsulation of siRNA into a photothermal PDA-coated nanocarrier for synergistic siRNA-mediated GT–PTT (Figure 15) [201]. The therapeutic complex consisted of a noncationic nucleic acid nanogel, in which anti-Hsp70 siRNA were fully embedded by mixing DNA-grafted polycaprolactone (DNA-g-PCL) and siRNA linker. To provide photothermal activity to the nanostructure, PEGylated PDA was linked to the surface through Michael addition. This multi-shielding enhanced the physiological stability, the blood circulation time and the accumulation at tumor site of the siRNA-bearing nanogel, protecting it from enzymatic RNase degradation and immune clearance. In contrast, these protections did not compromise the release of the functional siRNA, as upon cellular uptake the acidic environment of endocytosis-related intracellular organelles could provoke the degradation and detachment of the coating layer, revealing the nanogel to RNase H. Indeed, the efficient gene silencing against Hsp70, both in vivo and in vitro, made possible a low-temperature PTT, preventing the hyperthermia-induced side effects of traditional PTT. The achieved siRNA-mediated low-temperature PTT was able to completely suppress tumor growth upon laser irradiation in HeLa tumor-bearing mice model, as a result of the cooperative enhancement of the efficacy of PTT by GT.

In a similar way, Yang et al. demonstrated the activity of a melanin-poly-l-lysine (M-PLL) polymer as a siRNA vehicle, able to the overcome the endosomal barrier and to deliver into the cytoplasm the functional genetic material [202]. M-PLL nanoparticles exhibited good biocompatibility, biodegradability and siRNA electrostatic-binding capacity with an optimal ratio of 40:1 M-PLL/siRNA. Due to the high photothermal conversion efficacy of melanin, after the endocytosis of the flower-like shape NP by tumor cells, NIR irradiation was capable to induce local heat generation and further achieve “on demand” endosomal escape. This process resulted in enhanced gene silencing, which was proven by the use of luciferase-targeted siRNA. Furthermore, survivin, an inhibitor of apoptosis which plays a crucial role in breast cancer progression and metastasis, was exploited as a target-gene for anticancer therapy. Both in vivo and in vitro results, supported the excellent inhibitory effect on 4T1 tumor growth of the anti-survivin siRNA-loaded M-PLL. In a different approach, Mu et al. developed a cell membrane-coated PDA nanocarrier for enhanced target precision and siRNA delivery at tumor sites, coupled with imagine guided therapy [203]. The complex nanodrug, namely Fe_3_O_4_@PDA−siRNA@MSCs, consisted of an internal iron oxide-PDA core on which siRNA were bound to the PDA layer through π-π stacking and an external mesenchymal stem cells (MSCs) membrane vesicle coating. Due to the MSCs coating the complex NP displayed excellent tumor targeting ability as well as exceptional stability and biocompatibility. Moreover, in vitro results in DU145 cells verified the successful siRNA delivery, the excellent photothermal capability and MR-imaging functionality of the Fe_3_O_4_@PDA−siRNA@MSCs NP. In particular the authors focused on Plk1, which expression is elevated in tumor cells, showing that the delivery of siPlk1 through the nanovesicle was facilitated activating apoptotic pathways and inhibiting tumor cell growth as a result of the gene silencing. The synergistic combination of siRNA against Plk1 gene, which could inhibit the expression of the endogenous gene and cause apoptosis, with photothermal laser treatment showed clear antitumor efficacy in a DU145 xenograft mice model. According to the authors, considered, the significant inhibition of tumor growth in vivo the proposed NP represent a valuable alternative for imaging guided GT-PTT. Another possibility in GT, which was adopted by Fan et al., is the use of small and endogenous non-coding RNA, known as microRNA (miRNA) [204]. The aforementioned miRNA can control cancer-associated gene expression via the regulation of the level of messenger RNA or the suppression of the target mRNA translation. Correspondingly, poly-L-lysine functionalized MNP (MNP-PLL) were engineered for the delivery of mRNA with aim the treatment of laryngeal squamous cell carcinoma (LSCC). In order to obtain a stable nano-assembly, the positive charge of PLL was exploited to achieve the electrostatic adsorption of the negatively charged nucleic acids. Since the miR-145-5p expression is abnormally down- regulated in LSCC cell lines and tissues, the author used their miR-145-5p like conjugate to prevent the tumor cell migration and induce cell cycle arrest and apoptosis. In vitro and in vivo experiments were performed and the results demonstrated that the photothermal treatment associated to the miR-145-5p mediated GT was more efficient in promoting primary tumor suppression, compared to monotherapy. Additionally, this combination of therapies also inhibited the tumor progression eliminating the metastatic potential of tumor cells. Last but not least, the MNP-PLL nanoplatforms were adequate for photoacoustic imaging-guided therapy in vivo, something that conferred an additional advantage to this promising thermo-gene synergistic therapy. Non-viral gene delivery, constitutes another interesting possibility for GT. As an example, Zhang et al. constructed a melanin-based nanocarrier by covalently binding PDA with phenylboronic acid modified with polyethylene glycol (PEG-PBA) and with low-molecular weight polyethylenimine (PEI_1.8k_) [205]. In particular, boronic acids could form esters with PDA in alkaline conditions, obtaining pH dependent covalent bonds in the synthetized PDANP-PEI-rPEG NP. On the other hand, PEI was used in order to adsorb the desired gene, thanks to its proton sponge effect. In this way, the researchers obtained a stable and pH-dependent nanoplatform for gene delivery. Moreover, they showed, that the gene release could be triggered by NIR irradiation. In vivo results revealed an excellent photothermal conversion ability of the PDA-PEI-rPEG nanoparticles, something that suggested their feasibility in synergistic gene/PTT

#### 3.4.4. Chemo-Gene-PTT

Chemotherapy, GT and PTT can be readily integrated into a single nanoagent just by the co-encapsulation of an anti-cancer drug and genetic material into a photothermal conversion nanovesicle. One of the main advantages of the synergistic therapy is that the heat induced by PTT accelerates the drug release and also enhances the cellular uptake of the therapeutic agents [206]. Due to the exceptional performance as PTA along with their other intrinsic properties of biocompatibility and loading ability, the co-delivery of genes and chemotherapeutic drugs by MNP has shown great potential in the trimodal synergistic therapy. As an example, Cheng et al. designed a PDA-containing tumor-targeted nanoplatform for cooperative Chemo-Gene-PTT therapy that was made by a doxorubicin (DOX)-gated mesoporous silica nanocore (MSN) encapsulated with permeability glycoprotein (P-gp) siRNA into the pores and a PDA outer layer [207]. The melanin shell endowed the therapeutic system not only with excellent photothermal ability, but also with further modification capability by external ligands. Indeed, the outer PDA shell was exploited for the binding of folic acid and for the loading of DOX through π–π stacking and hydrogen bonding interactions. The robust theranostic agent was characterized by high cell-targeting selectivity and simultaneous effective delivery of both siRNA and DOX into MCF-7and MCF-7/ADR cells, supported by evidences indicating the effective suppression of P-gp protein on the cell’s surface. More precisely, DOX release was found to be both pH- and thermo-responsive, enhanced by PTT, with the release of the co-delivered siRNA to be pH-dependent. This outstanding photothermal activity of PTT, together with the thermo-pH-responsive drug and gene release, resulted in enhanced antitumor efficacy minimizing the potential damage to surrounded normal cells, in vivo. More recently Shim et al. constructed a number of different cargo-loadable DNA nanostructures, which were externally covered by PDA and additionally non-covalently tethered with nucleic acid aptamers to achieve cancer cell-specific targeting [208]. To begin with, rolling-circle amplification was applied for the preparation of the various DNA nanostructures, which were afterwards condensed by the use of adenovirus-derived Mu peptide, known to play a major role in condensation of viral genome. The versatility of these scaffolds was demonstrated by loading the DNA nanovesicle with antisense oligonucleotide, a photosensitizer or anticancer chemotherapeutic drug; hence the DNA nanostructure was afterwards externally shelled with PDA, and a further decoration with a poly adenine-tailed nucleic acid aptamer specific for PTK7 receptor, overexpressed in various cancer cells, was realized. The photothermal effect, induced by PTT, together with the surface modification with PTK-7-specific nucleic acid aptamers enhanced the cellular uptake and improved the delivery efficacy of the various anticancer therapeutic agents mentioned above. From all the other groups tested, DOX-loaded nanoplatforms exhibited the highest anticancer activity, as well as antisense oligonucleotide-loaded nanoplatforms, which provided selective reduction of target proteins. In a word, with the engineered complex nanodrug it was achieved specific receptor-targeted therapy, with the PDA coating to result in enhanced therapeutic activity by enabling the combination of active components delivery and PTT.

#### 3.4.5. Chemo-PTT- Immunotherapy

Some anticancer drugs, as well as some PTA, can act as immunologic adjuvants able to strengthen the cell-mediated immune response by promoting dendritic cell maturation and antitumor cytokine production [175]. With this in mind, a combined Chemo-PTT-immunotherapy strategy was reported by He and coworkers (Figure 16) who designed a novel nanoplatform based on hierarchical drug release, in order to achieve localized PTT, anticancer drug release and elicited immune response [209]. In particular, they started synthesizing biomimetic vesicles (BV) drug carriers via self-assembly strategy using cholesterol, soya lecithin and paclitaxel (PTX). PTX is widely used in chemo-immunotherapy as it acts as an immunologic adjuvant stimulating the immune system to eliminate residual tumors. These BV were demonstrated to be highly biocompatible and suitable to perform controllable drug release via temperature responsive phase transformation. In particular for photoactivation, BV were, in a first stage, modified by branched gold nano shells via seeded growth (BV/PTX@Au). Nevertheless, gold nanomaterials, that have been widely used in PTT, at the high concentration needed for therapeutic application, are known to produce toxic effects. For this reason, further modification with DOX loaded PDA encapsulation was carried out in order to increase biosafety, photothermal response and to add CT activity yielding the multicomponent system DOX/PDA@Au@BV/PTX who was demonstrated to perform a complex hierarchical drug release sequence. The mechanism could be described as follows: in first place, as result of tumoral cells endocytosis, the pH change triggered DOX release. In second place, local NIR irradiation resulted in heat production, achieving the PTT step. Finally, temperature increase caused PDA@Au@BV/PTX collapse, releasing PTX. The authors claimed that this combined strategy enhanced the therapeutic efficacy and inhibition of tumor recurrence.

### 3.5. Photodynamic Therapy (PDT) and Synergistic PDT-PTT

Photodynamic therapy (PDT) is a non-invasive, precise cancer therapy possessing several advantages in comparison to other cancer treatments. Very briefly, this therapy is based on the use of photo-sensitizers (PS) that when specifically excited generate cytotoxic ROS that cause cell death, general features of PDT have been discussed in recent review articles [210,211,212,213,214]. Considered this, the choice of the suitable light sources and the presence of oxygen in the target tissues are fundamental issues for the efficacy of the therapy. Moreover, as PDT relies on the cytotoxicity of intracellular ROS which are generated by the PS, in loco delivery of the photoactive compound at tumor regions and its effective internalization by cancer cells is needed [215]. On the contrary many PS show poor water solubility and nonspecific targeting [216] and the use of MNP for more selective PS deliverance has been proposed. In particular, three main strategies have been suggested to load the MNP with the PS: (i) binding covalently PS to MNP, (ii) encapsulating PS inside MNP, which act as both biocompatible coating and versatile nanoplatform and finally (iii) exploiting aromatic groups on MNP surface for PS adsorption. The last approach was found to be the simplest, but the PS-MNP conjugates resulting from surface modifications showed unsuitable circulation time and PS cells uptake. On the other hand, the method of encapsulation seemed to enhance PS stability, but, in this case, strong PS-MNP electronic interaction led to relevant PS excited-stated quenching, decreasing ROS production [217].

Poinard et al. suggested a two-step method, in which as PS, Chlorin e6 (Ce6) was loaded inside the polymeric matrix of PDA [218]. Chlorin e6 (Ce6) is a second-generation PS, which is widely used in PDT due to its high efficacy and low dark toxicity [219]. Since loading maintained the surface unaltered Ce6 containing MNP showed a potentially mucopenetrative enhanced performance, enabling more effective cancer phototherapy in organs, which require mucosal drug delivery. The nano-therapeutic agents exhibited prolonged release profile of the PS until day five leading to increased cell killing against T24 bladder cancer cells upon 665 nm laser irradiation.

As mentioned before, presence of oxygen is essential for PDT. Unfortunately, tumor tissue is characterized by extreme hypoxia, which leads to PDT resistance and finally diminishes the effectiveness of the treatment. Two strategies have been established in order to overcome this phenomenon: (i) the tumor-targeted oxygen-carrying and (ii) oxygen-producing. Considering this issue, a relevant application of PDA was proposed by Liu et al., who fabricated biomimetic aggressive man-made red blood cells (AmmRBC) for self-oxygen-supplied PDT (based on methylene blue as PS) against hypoxia-resistant tumor (Figure 17) [220]. For this application methylene blue-carried hemoglobin–PDA complexes were encapsulated inside nanovesicles composed by recombined RBC membrane. Importantly, AmmRBC, owning identical origin of outer membranes with RBCs, exhibited high biocompatibility, immune clearance escape capacity, effective tumor accumulation and, thanks to the oxygen transport action of hemoglobin, they could act as a self-oxygen-supplied nanoplatform able to overcome hypoxia-associated PDT resistance. PDA could protect oxygen-carrying hemoglobin from oxidation damage during the circulation.

Most recent application of MNP are focused on synergistic PTT-PDT rather than on simple PDT. This because MNP represent unique nanocarriers as they can load or encapsulate PS possessing also photothermal properties and excellent biocompatibility. As a synergistic effect, mild hyperthermia, induced by PTT, can increase the membrane permeability of the cells, leading to enhanced intracellular concentration of PS and thus enhanced ROS generation. Additionally, mild hyperthermia can accelerate the blood flow and meliorate tumor oxygenation, aiding in this way the oxygen-dependent PDT [175]. It is hence very interesting that PTT-PDT synergistic therapy can be, at least in principle, simply achieved by functionalizing MNP with PS. Also for this application surface control is important and Tian et al. adopted PEG modification to improve tumor-targeting ability, circulation time and biocompatibility of MNP [221]. Their PDMN–Ce6 NP were formed by the successful surface loading of Ce6 to PDA NP via π-π stacking, which were afterwards PEGylated. The formed NP exhibited as a whole preferential tumor accumulation and higher cellular internalization compared to free Ce6. Moreover, the in vivo results revealed that the synergistic PTT-PDT therapy significantly suppressed tumor growth and tumor angiogenesis with low off-target toxicity. In a similar manner, PEGylated PDA NP, chelated with iron ions and loaded with IR820 were reported to be suitable for PA/MR imaging-guided combined PTT-PDT [222]. Fe^3+^ ions acted as contrast agents for T1-weighted MR imaging and IR820, which is a NIR dye with a characteristic absorbance peak at approximately 820 nm, improved the NIR-responsive property of the NPs to enhance PA imaging, endowing them with dual-modal imaging capacity. What was significant about the NIR dye was that it enhanced the efficacy of PTT and additionally it induced efficient ROS generation, after cell internalization, activating PDT. A more synthetically complicated strategy was developed by Zhan et al., based on the concept of DNA complementary base pairing (in this case, adenine(A) and thymine(T)) (Figure 18) [120]. In this case, a thermolabile crosslinked PDA-zinc phthalocyanine (ZPC, widely used PS) complex was fabricated for cooperative PDT-PTT. More in depth, a photothermal-responsive PS-release nanosystem was developed through by binding adenine-modified PDA nanoparticles with thymine-modified zinc phthalocyanine PS (A-PDA = T-ZnPc). It is well known that, in DNA, heating induces the breakage of the base-paring in the double strand. This effect was exploited to induce the fracture of A-T hydrogen bonds and therefore release of zinc phthalocyanine upon laser irradiation of the NP at 808 nm because of the heat that photo-generated by PDA. Moreover, PDT was realized upon excitation of the PS at 665 nm, acting in a synergistic manner with the PTT-induced cell death for satisfactory anticancer results.

Single PDT was proven to exhibit low efficiency towards hypoxic solid tumors due to the, already mentioned, inadequate O_2_ supply in tumor vascular systems. In order to overcome this obstacle a core-shell photothermal agent was designed, which could achieve tumor-specific drug administration and simultaneous hypoxia amelioration via sufficient O_2_ generation from endogenous hydrogen peroxide. Illustratively, the PS methylene blue (MB) and a catalase (CAT) were simultaneously delivered into tumor cells via their encapsulation in ZIF-8-gated PDA nano-carrier [223]. The ZIF-8 shell acted as a smart gatekeeper facilitating the simultaneous and also effective delivery of the two payloads into tumor tissues, where the acidic tumor microenvironment could trigger their slow release via acidic dissolution of ZIF-8. This pH activated release allowed to prevent the premature MB exposure to blood or normal physiological tissues. The potency of the nanoplatform for PDT-PTT synergistic therapy was demonstrated both in vivo and in vitro, with a key point to be the CAT-mediated self-sufficient O_2_ generation for PDT operation, acting alongside with thermally-induced cancer cell killing.

### 3.6. Antibacterial Infections

Even if antibiotic treatment has achieved significant results against various bacterial infections many antibiotics, like Gentamicin, present low bioavailability and cause serious side effects. Exploiting the high biocompatibility and easy drug-loading capacity of MNP, one possible therapeutic approach, able to improve the therapeutic efficacy and to reduce the side-effects associated with gentamicin, was to utilize MNP as antibiotic nano-carrier. Indeed, by this method MNP can act as a gentamicin nanoreservoir and when the amount of drug loaded is maximum the inhibitory concentration and bactericidal concentration are minimum [224]. On the other hand, antibiotic resistant bacteria spread is rising at an alarming rate, thus finding new strategies and potent antibacterial materials has become demanding [225]. Starting with the simplest approaches, MNP were chelated with specific metals that are known for their antimicrobial properties. Using facile one-pot sonochemical synthesis, Yeroslavsky et al. fabricated core-shell Cu-, Ag-, and hybrid Cu/Ag-based PDA nanoparticles [226]. The formed NP exhibited broad and more potent antibacterial activity than commercial Ag-NP, as they were also effective against robust biofilms. To be more specific the combination of both metals was found to be the most potent acting also in combination with semiquinone, generated by PDA, that also contributed in the antibacterial action. In a different approach, PDA hollow NP functionalized with N-diazeniumdiolates as a nitric oxide (NO) delivery nanocarriers were proposed as a metal-free antibacterial alternative [227]. NO antibacterial system is based on the induction of oxidative stress or nitrosative stress able to kill various kind of bacteria. In fact, the developed NP showed sufficient NO release and excellent bacterial killing ability against Gram-negative bacteria. As non-invasive techniques are gaining more and more attention, recent antibiotic NP exploited also the intrinsic property of MNP to act as a PTA. In fact, PTT can also act synergistically with antimicrobial-NP in order to inhibit bacterial growth, disrupt their membrane and offer an enhanced antibacterial effect [228]. Taking this into account, Li et al., reported a novel PDA composites that showed promising antibacterial properties against methicillin-resistant *Staphylococcus aureus* (MRSA) [60]. MRSA eradication was achieved thanks to the synergistic effect of PTT and Lysozyme (Lyso) assisted antibiotic action [229]. Lyso is an antimicrobial enzyme that catalyzes the hydrolysis of peptidoglycans, the major components of Gram-positive bacteria cell wall. PDA-Lyso conjugation was possible thanks to electrostatic interactions, since Lyso surface is positively charged. A “Lyso-assisted phototherapy” mechanism was proposed by authors: local heat produced by PDA NIR irradiation was demonstrated to damage the cells membrane. Since then, Lyso-bacteria interaction has been favored, leading to the complete destruction of membrane. Completely different approach was proposed by Liu et al. who developed a novel pH/near-infrared (NIR)-responsive hydrogel for on-demand drug delivery and wound healing based on the crosslinking of polymeric network and NP [146]. In particular, hydrogel was synthesized via Ca^2+^ physical cross-linking of PDA and cellulose nano-fibrils, which were produced by 2, 2, 6, 6-tetramethylpiperidine-1-oxyl (TEMPO)-mediated oxidation method. Excellent pH/NIR responsive ability has been shown under lower pH value or NIR irradiation and an “on-off” pattern resulted from NIR laser irradiation. Moreover, drug release was proved to be continuous over 24 h without NIR laser irradiation. Hydrogel was successfully applied to skin wound dressing test, and it showed good antibacterial properties against *Staphylococcus aureus* and *Escherichia coli* [230].

## 4. MNP and Nanocosmetics

Skin pigmentation is of great cultural and cosmetic importance [231]. Human skin is repeatedly exposed to UV radiation that influences the function and survival of many cell types and that is regarded as a major cause in the initiation of skin cancer. Sunlight, in particular, stimulates melanocytes, specific dendritic cells, to synthesize melanin, that is then transferred into the surrounding keratinocytes by melanosomes [232]. Skin pigmentation is one of the most important photoprotective factor, as melanin, except for acting as broadband UV absorbent, has additional AOX and radical scavenging properties [32]. According to Solano et al., for photo-protection, it is essential that melanin releases the absorbed energy through ultrafast thermal processes without producing any long-lived, possibly reactive, transient species [77]. If in a first stage the natural properties of melanin inspired its integration in artificial structures like MNP, more recently, the possibility of use these nanostructures as replacement of natural melanin started to be considered. This is due to the fact that, for example, often the natural photo-protective action of the skin is not sufficient and it needs to be supplemented by the use of synthetic solar filters. In another important case the natural coloring effect of natural melanin in human hair can fade as a result of aging inducing a culturally accepted need of restoration. Both issues are related more to cosmetics rather than to medicine and we can say that melanin in nature plays a cosmetic function. It is quite straightforward then to consider that MNP can find application in cosmetic formulation, as it will be shown in subsequent sections, paving the road to the birth of a new science that we can call nanocosmetics.

### 4.1. MNP for Photoprotection

Knowing, that the amount and the type of melanin in the human epidermis, thus the skin color, is genetically determined [233], scientist tried to mimic the behavior of the natural pigment and exploit the ubiquitous properties of different synthetic MNP for photoprotection and cosmetic application. To begin with, Vij et al. used a chemo-enzymatic method to prepare peptide-derived MNP [234]. The aforesaid synthesis led to different color shades synthetic melanin variants, which were evaluated for their photoprotective activity in human skin derived keratinocyte HaCaT cells, measuring the sun protective factor (SPF). It was observed that the cellular uptake of the NP increased proportionally with increasing the incubation period and that most of them were located at the nuclear periphery or above the nucleus. This feature mimics the tendency of natural melanin to form a protective covering around the nucleus of the skin cells in order to prevent UV-induced damage of their genetic material, DNA. Due to their efficient UV-protection ability, their low cytotoxicity, high durability and different color perception, these NP serve a promising innovation in the cosmetic industry to achieve the desired skin color tone along with photoprotection, as they fulfill many of requirements that a UV-filter possesses, including the ability to be stored at ambient conditions without significant alteration.

This is not the only example in which melanin was a source of inspiration for the development of different colors. For instance, Xiao et al. mimicking natural melanosomes in structurally colored feathers, generated colored films by the deposition of PDA NP in thin film [235]. In a later study, by Huang et al., it was demonstrated, from stem to stern, that the pathway of transportation and lysosomal degradation of artificial MNPs in HEKaT cells was identical to the pathway of natural melanosomes [236]. More precisely, PDA NP, produced by spontaneous oxidation of dopamine in alkaline solution, can mimic the behavior of natural melanosomes and protect the cells from UV damage through endocytosis, accumulation and formation the so-called micro parasols or perinuclear caps. In addition, the photoprotective activity of PDA NP against UV-induced radicals, which is related to their AOX property, was verified by using a ROS-activated fluorescent marker, DCFH-DA. As expected, the amount of ROS was significantly lower in cells treated with the MNP than this observed in the untreated cells. To further expand the photoprotective properties of the bioinspired melanin NP, Zhou et al. proposed the synthetic development of the fungi-derived nitrogen-free melanin, allomelanin (Figure 19) [58]. Different allomelanin-like NP were synthetized by oxidative oligomerization of 1,8-DHN in aqueous solution at room temperature. Despite the different chemistry of the precursor, the absence of nitrogen and the fact that this kind of melanin is not naturally found in humans, the NP surface chemistry played the governing role in the cellular trafficking. Indeed, the incorporation of the artificial allomelanin NP in human epidermal keratinocytes and the formation of microparasols was consistent with the anterior studies. Above all, the feature that discriminated this kind of MNP was their exceptional radical scavenging activity. To be more precise, the allomelanin-like NPs were found to have AOX activity similar to ascorbic acid and better than the previously investigated PDA NP. The previous studies clearly demonstrated the utility of the different types of MNP in sunscreens and protective coatings. Combining the fascinating photoprotective properties of PDA with a cosmetic base, a skin-pigmentation inspired PDA sunscreen was designed by Wang et al. [237]. Illustratively, PDA NP were incorporated in three widely cosmetically used polymers, creating sunscreen gels with superior UV shielding properties, high in vitro and in vivo UV protection efficiencies, no phototoxicity, and nonirritating nature. What was unique about these melanin-like sunscreens was their sufficient skin retention ability, as their adhesion on skins was quite stable with additional water-resistant nature. On the other hand, the gels did no penetrate on the skin and could be mechanically removed by towel wiping. MNP were demonstrated to be useful not only UV-Vis photoprotection, in fact, novel radical-enriched MNP has been reported to be suitable even for protection against ionizing radiation [35]. The introduction of stable nitrogen radical in the synthetic PDA nanoparticles significantly enhanced the X-ray protection properties of the MNP, which exhibited different electrochemical and magnetic properties in comparison to conventional PDA. Importantly, the NP were successfully internalized by human epidermal keratinocytes, mimicking the behavior of natural melanosome, where they significantly eliminate X-ray-induced ROS. In short, the possibility to produce simple, safe and efficient bioinspired sunscreens for human skin protection, covering a wide range of the electromagnetic radiation, envisages the continuation of relevant experimentation on the cosmetic and pharmaceutical application of MNP.

### 4.2. Hair Coloration

Human hair color is determined by the mixture of the black-brown eumelanin and the red-yellow pheomelanin [238]. The regulation of hair pigmentation is affected by numerous intrinsic factors including general metabolism, racial or gender differences and variable hormonal influences. Even so, hair graying is related closely with chronological aging and occurs in all individuals, regardless of gender or race [239]. Taking into consideration that conventional oxidative hair dyes contain ingredients as para-phenylenediamine, para-toluenediamine, substituted para-diamines etc., which were recognized by several studies as potent carcinogens and allergens [240,241], some researchers proposed for coloration completely new materials, such as, for example, graphene [242] despite its actual biocompatibility is still debated [243]. Regarding melanin, being naturally responsible for hair coloration of course, its restoration is the most attractive strategy for hair dyeing. Nevertheless, this achievement is far from being trivial. Im et al. [244] suggested the use of PDA NP as a substitute of permanent hair dyes. PDA, produced by oxidative polymerization, in combination with ferrous ions (Fe^2+^) was capable to accomplish permanent coloration of gray hair in a relatively small time period (1 h). An important feature of the bioinspired dye was to be resistant to detergents. On the other hand, PDA alone provided an insufficient extend of coloration. In addition, a variety of colored dyes was created by simply changing the kind of metal ion used. Following this, Feng Cao et al. reported a PDA-based hair dye, that achieved PDA deposition on hair surfaces using copper ions and hydrogen peroxide (H_2_O_2_) as a trigger [245]. CuSO_4_ and H_2_O_2_ could rapidly induce the deposition of the PDA coating onto the hair surface in almost 5 min, a time much faster than commercially found hair dyes. It should be also highlighted that this innovative hair dye possessed three additional features: strong adhesion capacity, as it faded only slightly after 30 washes, thermal insulation performance and incredible antibacterial performance. Differently from the previous approaches, Battistella et al. suggested a metal-free PDA-based long-lasting hair coloration method was, which was not only cost-efficient, but also was carried out under mild conditions [246]. According to the study, PDA was efficiently deposed on human hair without the need of metal chelators or strong oxidants and different colors were achieved just by tuning the reaction conditions. These authors demonstrated that hair coloration can be achieved efficiently in 2 h simply by raising the temperature to the physiological values of 37−40 °C, as with the increase of temperature, the polymerization of dopamine was more efficient, and a dark and uniform hair coloration was achieved in a relatively short period. Additionally, the increase of the base concentration, up to 6%, resulted in even a darker color, whereas the addition of H_2_O_2_ led to warmer and orange/gold shades (Figure 20). According to the authors, owing to the mild and inexpensive conditions employed, this novel approach has the potential to replace classical harsh hair dyeing conditions that have raised concerns for several decades due to their potential toxicity.

## 5. Conclusions and Perspectives

Recent developments of bio-application of MNP to nanomedicine and more recently to nanocosmetics reveal the unique versatility of both natural and artificial melanin for the design of multifunctional nanoplatforms. The variety of properties of these bio-inspired and intrinsically biocompatible materials are, as widely discussed in the previous sections, exceptional. Thanks to the broad band absorption and ultra-fast excited state thermal deactivation, MNP are intrinsic imaging contrast agents for PAI and simultaneously efficient PTA for PTT. Intrinsic fluorescence of MNP, even if weak, can also be exploited for imaging expecially if supplemented by functionalization with additional fluorophores. Functionalizability is more in general a key feature of MNP that can be simply converted into contrast agents for MRI by metal ion complexation. They become nanocarriers upon loading with drugs for CT or other therapeutic agents for immuno-T or gene-T or even PS for PDT. The surface of MNP can be easily modified with “stealth”-polymers or with specific ligands for targeted recognition. Even more interestingly, all these possibilities can be easily combined to achieve multi-modal imaging and multi-therapeutic action. The stimuli induced responses can be interconnected in order to couple different functions to achieve e.g., controlled drug release upon light stimulation via photothermal effect, producing in the simultaneously PA signal and localized PTT. The variety of possible operative schemes offers, for MNP, unprecedented possibilities and an almost infinite sequence of approaches. Nevertheless, although from the applicative point of view, obtaining an optimally performing theranostic nanoplatform is a major issue in nanomedicine, a complete understanding of the inner nature and of the chemical, photophysical and photochemical properties of the wide range of materials classified as melanin, that constitute MNP, is still missing. The same chemical composition and process of formation vary from system to system and they can be only in part generalized since indeed, they depend, to a large extent, on the preparative conditions and on the presence of possible co-reactants. These differences are mirrored by the modification of properties as demonstrated by the well documented difference between eumelanin, that is known to play an antioxidant function in nature, and pheomelanin that has been reported to be pro-oxidant. In general, in most of the important examples reported in the scientific literature and discussed in this review article, melanin is incorporated in the nano-agents, that we generally indicated as MNP, by adapting well-established generic procedure without considering in detail the effect of the changed synthetic conditions and of the specific variations on the final properties. As a consequence, in many cases, it is very difficult to extract information about the actual composition of the resulting melanin-like material. This lack of information arises in part from the undoubtable technical difficulty found in characterizing nanomaterial composition especially in the case of heterogeneous and multi-structured NP. Nevertheless, we believe this is a major weakness of the quite generalized approach to the design of melanin containing nanomaterials that do not consider adequately the variability in terms of structure and properties of melanin itself. We also believe that a more accurate control of the melanin formation process may lead, in the future, also to best performing theranostic NP. In conclusion, a more detailed comprehension of the chemical properties of the different kind of synthetic melanin is necessary, as well as a more systematic classification of the different types of materials, an issue that recently was faced for other highly promising and exploited carbon-based nanostructures like graphene and related materials [247]. Many experimental evidences proved the coexistence in melanin of covalent and non-covalent interactions demonstrating that this is, to a large extent, a supramolecular hierarchical assembly. Together with the possible partial reversibility of the polymerization reaction, non-covalent interactions make melanin an extremely dynamic structure that responds in multiple-ways to changes in concentration or, considered the presence of protonable/deprotonable groups, to pH variations. This responsivity is surely highly promising in nanomedicine but, again, the actual physico-chemical changes produced by stimulation would deserve deeper investigation. In this context, optical techniques, e.g., ultrafast transient absorption, offer great advantages allowing to analyze and distinguish in real time the different components of this promising but complicated and heterogeneous material.

As far as bio-applications are concerned, across all this review paper we have underlined as the main driving force for the use of melanin related materials in nanomedicine is their intrinsic biocompatibility. Indeed, also in this case, the poorly specific classification of melanin derivatives may generate confusion. The existence of eumelanin in human tissues and its recognized function as natural pigment and photoprotective agent is not itself a definitive demonstration of the safety of any similar natural or artificial substance. Indeed, as also recently underlined by Liu et al. [47], the biodistribution, biodegradation, and metabolic pathway of melanin related nanomaterials need to be investigated case by case, considering, as a possible set of parameters, the actual source, method of synthesis or extraction, preservation, and formulation. Without these systematic studies, potential mid- and long-term toxicity on organs of at least some of the NP classified as melanin cannot be definitively excluded. We would also add that most of the NP system described here for in-vivo application have been demonstrated as effective on an animal model, but actual pharmacokinetic and penetration and accumulation in target tissues in the human body need to be demonstrated for translation at the clinical level.

Despite further research is needed, MNP and related materials, because of their multi-faceted properties and the easiness of application, will surely have a bright future in nanomedicine and in the newborn science of nanocosmetics.

## Figures and Tables

**Figure 1 nanomaterials-10-02276-f001:**
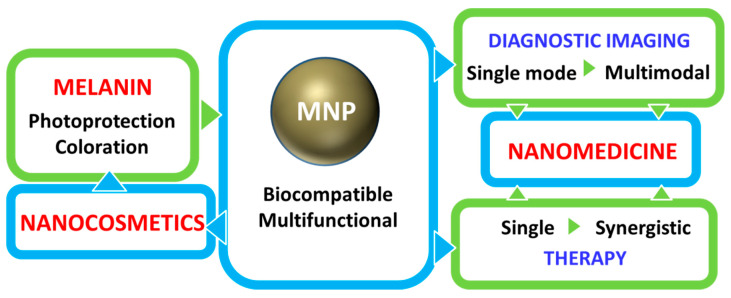
Scheme summarizing the interconnections between melanin, MNP and their applications. Melanin in nature is responsible for photoprotection and coloration. Its properties inspired the development of MNP for imaging and therapy and finally for theranostics in nanomedicine. Recently MNP have been considered as a supplement to natural melanin for photoprotection and coloration contributing to the development of nanocosmetics.

**Figure 2 nanomaterials-10-02276-f002:**
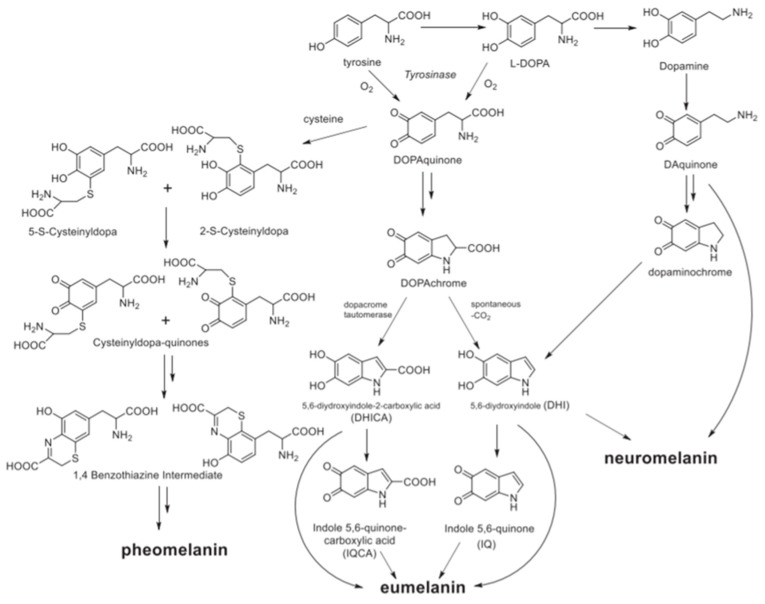
Reactions involved in the formation of natural pheomelanin, eumelanin and neuromelanin starting from tyrosine.

**Figure 3 nanomaterials-10-02276-f003:**
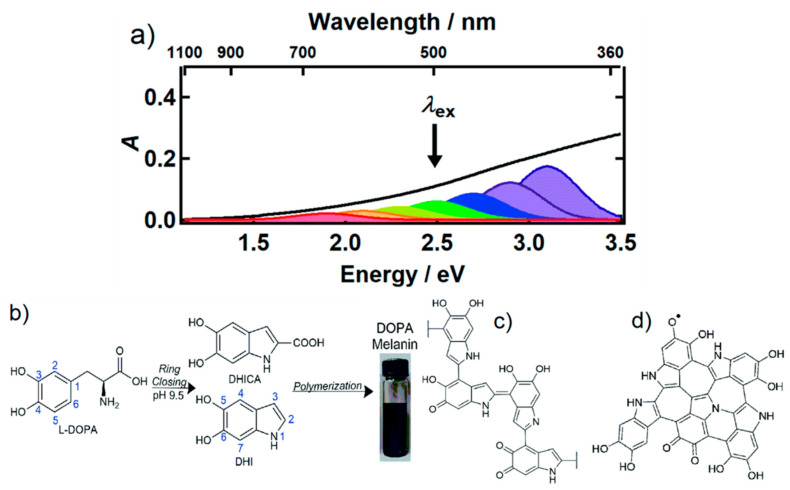
(**a**) UV-Vis-NIR absorption spectrum of DOPA melanin in aqueous solution (black line). Hypothetical distinct chromophores, some of which can be selectively excited at selected wavelength (black arrow), are shown as colored Gaussian functions. (**b**) Synthesis of DOPA melanin: L-DOPA oxidation produces DHI and DHICA intermediates that polymerize to give the typical structural motifs proposed for eumelanin shown in (**c**) and (**d**). Adapted from [69], with permission from Royal Society of Chemistry, 2020.

**Figure 4 nanomaterials-10-02276-f004:**
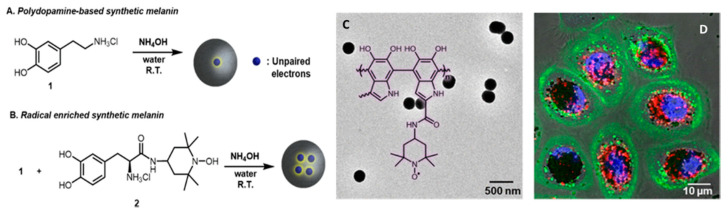
(**A**) Synthesis of polydopamine-based NP. (**B**) Radical-enriched MNP result from the copolymerization of dopamine with radical monomers generates. (**C**) TEM images of the NP. (**D**) Confocal microscopy of NHEK live-cells after incubating radical NPs. Adapted from [35], with permission from American Chemical Society, 2020.

**Figure 5 nanomaterials-10-02276-f005:**
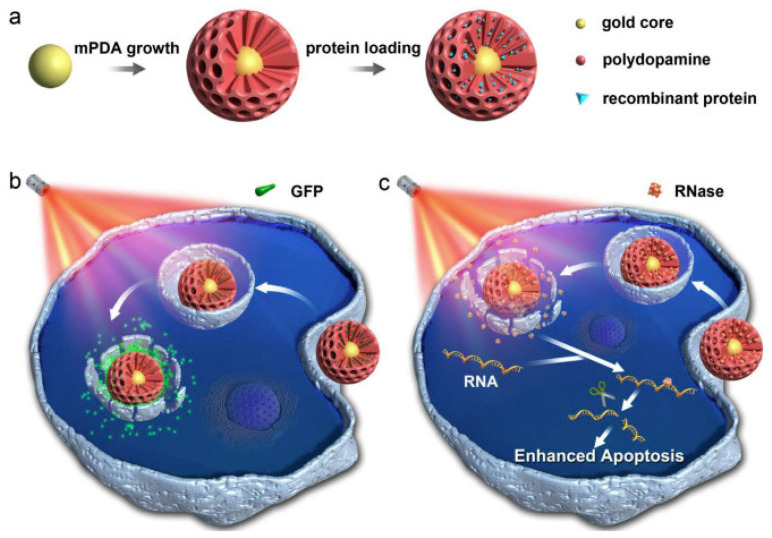
(**a**) Au@mPDA loading with protein. Intracellular delivery and release of (**b**) GFP and (**c**) RNaseproteins activated by NIR irradiation. Reproduced from [108], with permission from Elsevier, 2020.

**Figure 6 nanomaterials-10-02276-f006:**
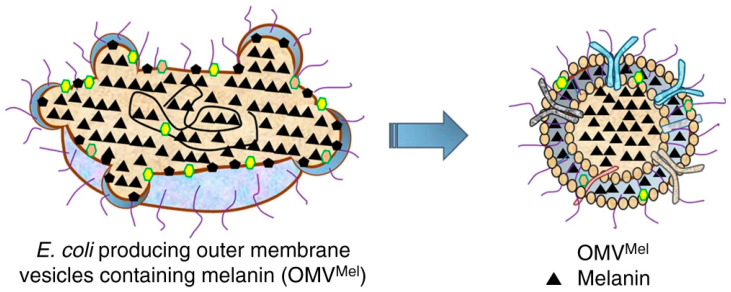
Scheme of the formation of OMVMel (OMV is outer membrane vesicle): OMVMel is purified after vesiculation from the parental bacteria. Reproduced from [116], with permission from Springer Nature, 2019.

**Figure 7 nanomaterials-10-02276-f007:**
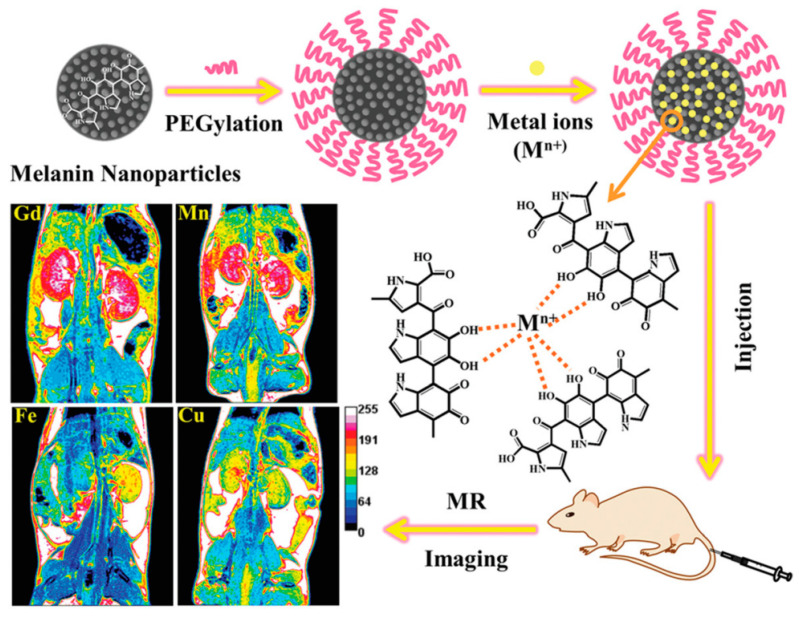
Scheme of the synthesis of MNP loaded with metal ions proposed as MRI contrast agents. Reproduced from [36], with permission from Royal Society of Chemistry, 2020.

**Figure 8 nanomaterials-10-02276-f008:**
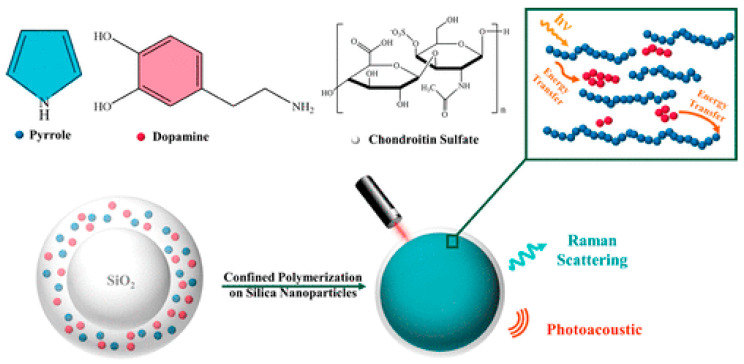
Scheme of the formation of PPy–PDA Hybrid Nanoshells. The energy transfer process between the components is also schematized. Reproduced from [135], with permission from American Chemical Society, 2018.

**Figure 9 nanomaterials-10-02276-f009:**
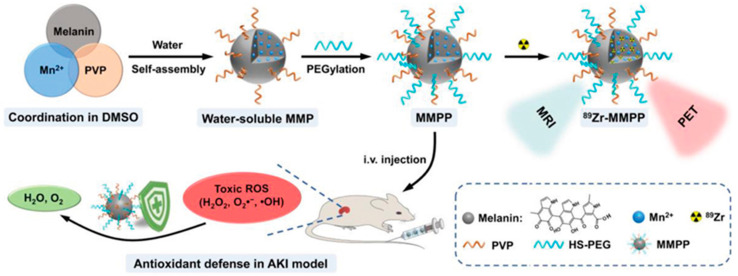
Scheme of the synthesis of MMPP for PET/MR bimodal imaging-guided AKI therapy. The activity of the NP as a naturally antioxidative platform is also schematized. Adapted from [140], with permission from John Wiley and Sons, 2019.

**Figure 10 nanomaterials-10-02276-f010:**
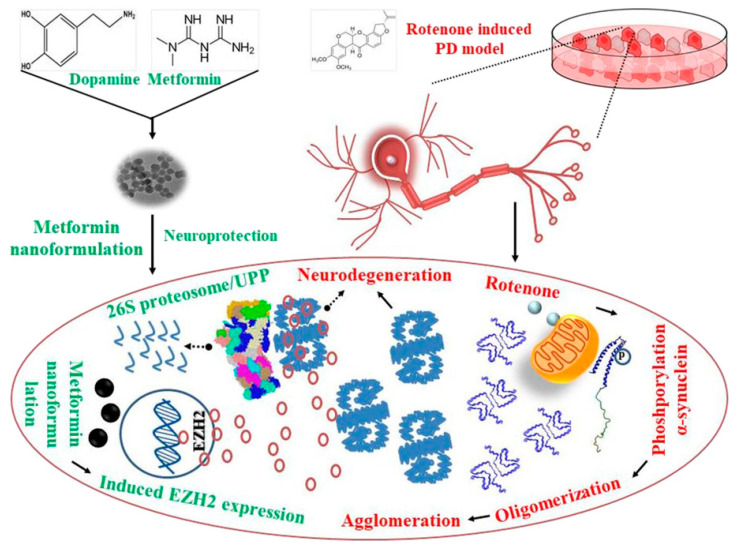
Scheme showing the neuroprotective action of Metformin nanoformulation (Met loaded PDANPs). Resulting NP inhibit rotenone-induced neurotoxicity by promoting EZH2 mediated α-Synuclein ubiquitination and degradation to prevent PD pathogenesis. Reproduced from [147], with permission from Elsevier, 2020.

**Figure 11 nanomaterials-10-02276-f011:**
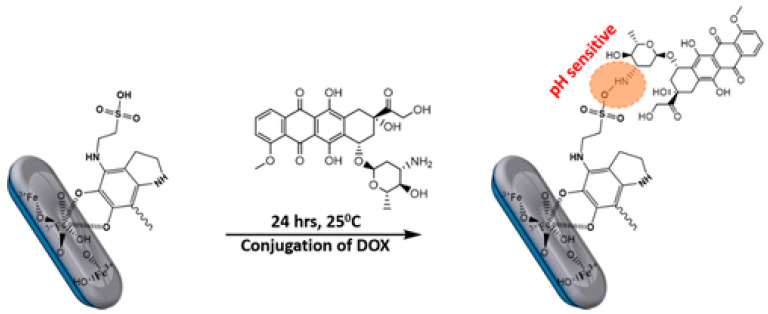
Scheme of preparation of doxorubicin/T-pDA-Fe_3_O_4_ NP conjugates. Sulfonamide link is exploited as pH-responsive group to activate the release of drug. Reproduced from [153], with permission from American Chemical Society, 2020.

**Figure 12 nanomaterials-10-02276-f012:**
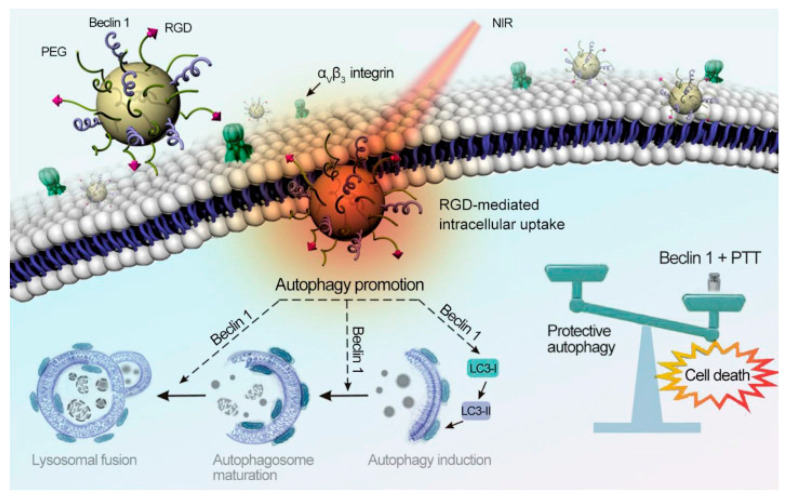
Beclin 1-induced autophagy was exploited for photothermal killing of cancer cells. Internalization of PPBR by cancer cells, via RGD-αvβ3 recognition, was exploited while Beclin 1, bound to the NP surface, contributed to up-regulate autophagy to sensitize cancer cells to PTT. Reproduced from [171], with permission from Elsevier, 2019.

**Figure 13 nanomaterials-10-02276-f013:**
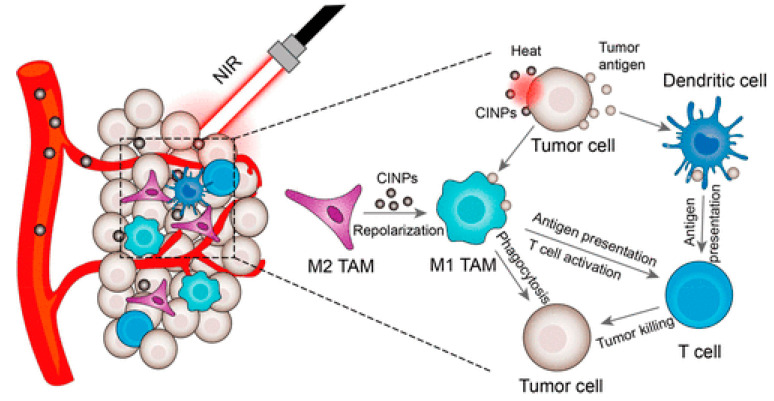
Scheme showing the role of CINPs in inhibiting tumor growth by producing macrophage repolarization and activating synergistic photothermal therapy. Reproduced from [181], with permission from American Chemical Society, 2019.

**Figure 14 nanomaterials-10-02276-f014:**
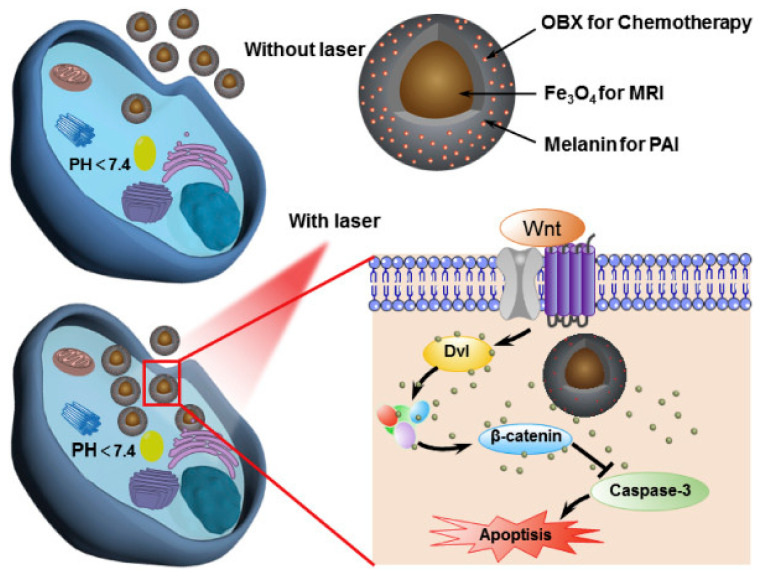
Structure of the OBX loaded MMNs (OBX-MMNs) for PA/MR multimodality imaging guided mild hyperthermia-enhanced chemotherapy. OBX is exploited as an inhibitor of the Wnt/β-catenin signaling pathway. Reproduced from [194], with permission from Elsevier, 2020.

**Figure 15 nanomaterials-10-02276-f015:**
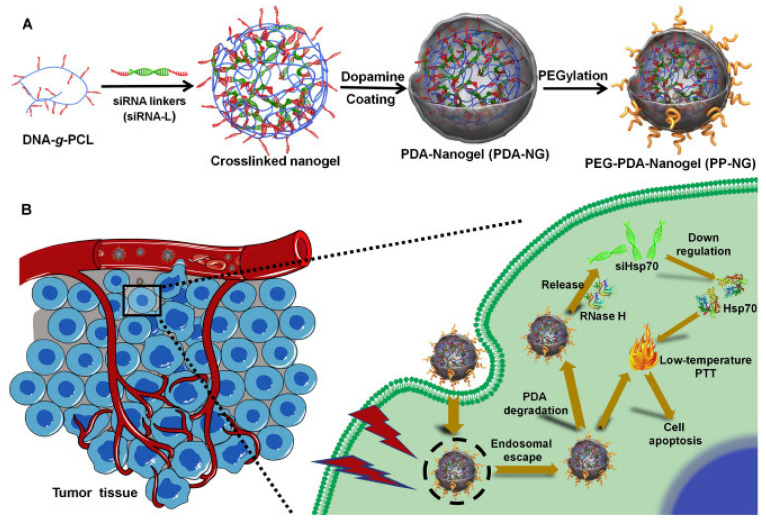
Scheme of the preparation of PDA-coated nucleic acid nanogel and its application in siRNA-mediated low-temperature PTT in vivo. (**A**) Synthesis of PDA-coated nucleic acid nanogel with PEGylated surface (PEG-PDA-Nanogel). (**B**) Processes involved in the siRNA-mediated low-temperature photothermal therapy induced by PEG-PDA-Nanogel. Reproduced from [201], with permission from Elsevier, 2020.

**Figure 16 nanomaterials-10-02276-f016:**
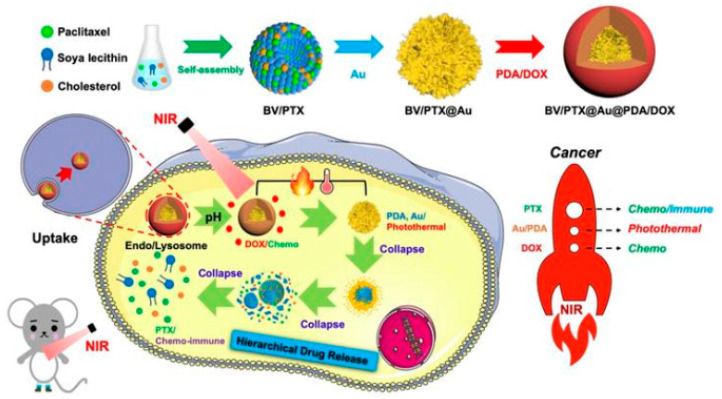
Schematic illustration of the preparation of BV/PTX@Au@PDA/DOX NP (top) and of the hierarchical drug release process (bottom) that combine pH-activated and NIR-triggered localized/systematic cascade cancer therapy in tumor-bearing mice. Reproduced from [209], with permission from Ivyspring International Publisher, 2019.

**Figure 17 nanomaterials-10-02276-f017:**
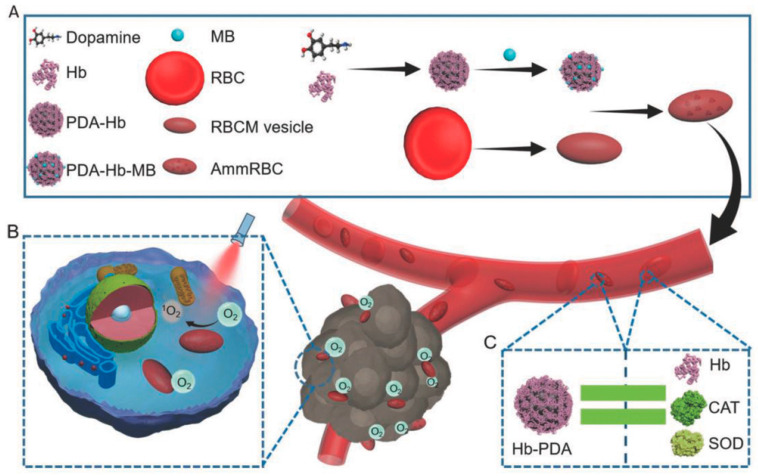
Scheme of the preparation and application of AmmRBCs for tumor therapy. (**A**) Preparation of AmmRBC. (**B**) Accumulation of AmmRBCs in tumor site enhances singlet oxygen generation for more efficient PDT. (**C**) PDA in AmmRBC mimics CAT and SOD in RBC to protect Hb from oxidant damage during the circulation. Reproduced from [220], with permission from John Wiley and Sons, 2018.

**Figure 18 nanomaterials-10-02276-f018:**
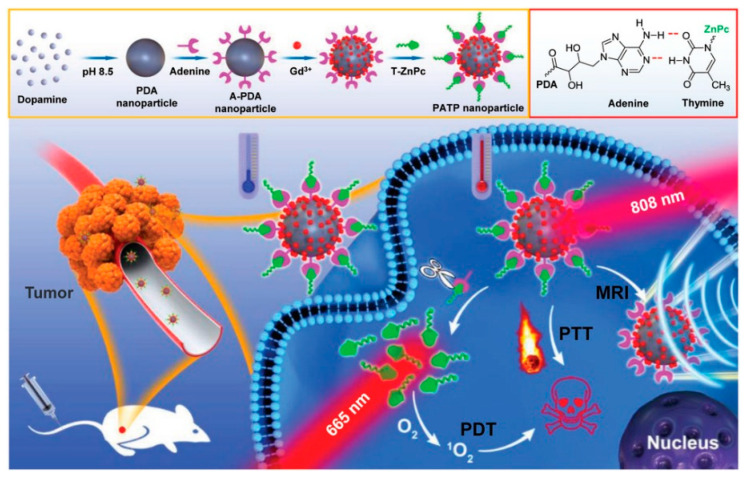
Schematic illustration showing synthesis, delivery, and the process of T-ZnPc release from PATP NP for its PTT/PDT treatment to tumor. Reproduced from [120], with permission from John Wiley and Sons, 2019.

**Figure 19 nanomaterials-10-02276-f019:**
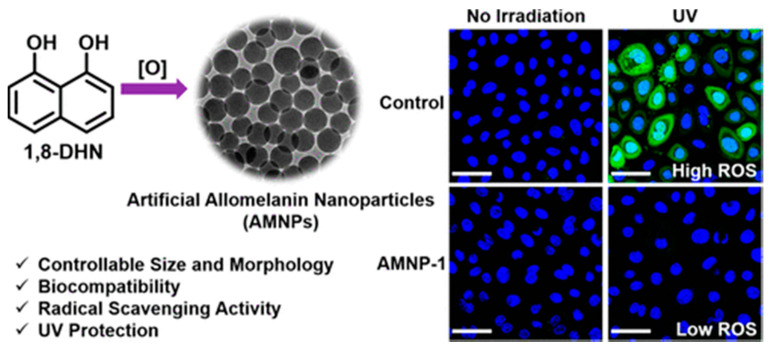
Schematic illustration of the preparation of allomelanin NP. Confocal microscopy images of NHEK cells incubated the NP are also shown. Reproduced from [58], with permission from American Chemical Society, 2019.

**Figure 20 nanomaterials-10-02276-f020:**
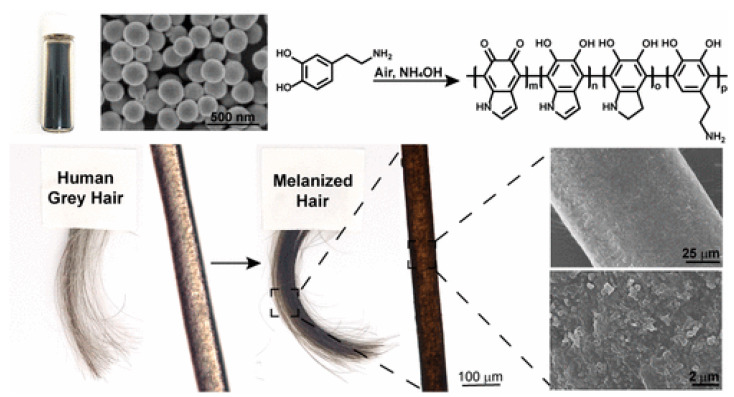
Schematic illustration of the synthesis of PDA NP and optical and SEM images of hair before and after treatment with the NP. Reproduced from [246], with permission from American Chemical Society, 2020.

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
