# Peer review of "Bio-Applications of Multifunctional Melanin Nanoparticles: From Nanomedicine to Nanocosmetics"

_nanomaterials, 2020, doi:10.3390/nano10112276_

Round 1
Reviewer 1 Report
The manuscript by Mavridi-Printezi et al. reviews the development and perspective of multifunctional melanin nanoparticles (MNPs). The review presents a good overview of the history, the principle behind the synthesis and properties, and the selected applications including mono/multimodal imaging, photothermal/dynamic therapies, theranostics, and nano-cosmetics. I think that the review is well written and informative; therefore, I recommend this work for publication with some minor suggestions.
1. Since this review consists of a lot of sections and subsections, providing a table of contents (if available) may help readers to overview the contents and organization of the manuscript.
2. Minor spell/spacing check would be required.
Reviewer 2 Report
[General comments]
The authors reviewed the recent literature on applications on natural and artificial melanin nanoparticles in medicine and cosmetics. They start from discussing physicochemical properties of melanin as well as melanin-based nanoparticles. Further, they perform a summary of recent development in using this nanopigment and its derivatives in various applications: medical imaging, diagnostics, and therapies. Finally, authors provide a relatively short overview of applications of melanin nanoparticles in cosmetics.
There are four major problems of this contribution:
- The authors copied whole sentences from original papers of other authors. Even if correctly cited (not always the case, see [Detailed comments]), these practices are unacceptable. Copying a portion of text from another source without giving credit to its author and without enclosing the borrowed text in quotation marks is a definition of word-for-word plagiarism (Roig, Miguel. (2011). Avoiding plagiarism, self-plagiarism, and other questionable writing practices: A guide to ethical writing.). Moreover, the authors borrowed most of the figures in the article (including captions) without reference to the corresponding permission from the publisher.
- A good review article needs to go beyond the recent literature summary. The authors only occasionally are trying to challenge the ideas of the work they review. The authors do not guide the reader in understanding the importance of discussed topics by providing their own perspective in the broader context, and rarely question the value of the conclusions made in the reviewed literature.
- Similar review papers have been recently published. The motivations of publishing another contribution of analogues content are missing.
- Liu et al. Melanin‐Like Nanomaterials for Advanced Biomedical Applications: A Versatile Platform with Extraordinary Promise. Sci. 2020, 7, 1903129.
- Zheng-Yuan Hong et al. Melanin-based nanomaterials: The promising nanoplatforms for cancer diagnosis and therapy, Nanomedicine: Nanotechnology, Biology and Medicine, Volume 28, 2020, 102211.
- Qi C, et al. Melanin/polydopamine-based nanomaterials for biomedical applications. Sci China Chem, 2019, 62:162–188.
- Park et al. Recent advances in melanin-like nanomaterials in biomedical applications: a mini review. Biomaterials Research, 2019, 23-24
- The questionable quality of the literature research throughout the manuscript.
A characteristic example can be found in lines 1513-1515. The authors state: “Photodynamic therapy (PDT) is a non‐invasive, precise cancer therapy possessing several advantages in comparison to other cancer treatments. In particular, PDT does not cause long‐lasting side‐effect, it is cost-efficient and can be applied multiple times.” There is only one paper cited, where NIR absorbing dyes for Photodynamic therapy were reviewed and photodynamic mechanism of these dyes were covered. This paper does not discuss or review the role, strengths and weaknesses of PDT in the cancer therapy.
[Detailed comments]
- Word-for-word plagiarism examples: 57-58, 62-63, 82-83, 93-94, 200-206 (whole passage, partially copy-pasted from Karlsson et al. [125], and not cited), 292-300 (whole passage), 1420-1426 (whole passage), 1648-1650, 1653-1654.
Captions of figures, e.g.: Fig. 3, Fig. 6, Fig 7 or Fig. 16 copied 1:1 from original papers.
- 95-97: “melanin […] as nanosized particles, the melanosomes […] as nanopigments”. The sentence might be misleading suggesting that melanosomes are nothing else than pigment, which is obviously not true. Melanosomes harbour numerous tissue-specific proteins such as Tyrosinase, TRP1 or TRP2. In fact, immature melanosomes (Stage 1 and Stage 2) lack pigment. Interestingly, pheomelanosomes differ in structure and composition from melanosomes harbouring eumelanin.
- 137: van der Waals interactions
- 180-190: “In nature, brown‐black eumelanin and red‐yellow pheomelanin are derived from tyrosine precursor, and their main difference is whether they contain elemental sulfur or not. […] In the case of pheomelanin the main difference is the presence of sulfur in the aromatic ring structures of the bio‐pigment.” This sentence includes imprecise and redundant information. Eumelanin polymers are composed of DHI and DHICA units, while pheomelanin polymers are composed of benzothiazole and benzothiazine units as the authors themselves correctly stated later 196-197.
- 200: Would be beneficial to comment on the phototoxic properties of pheomelanin and provide more literature. Initial and delayed CPDs were more frequent in mice whose melanocytes synthesized red-yellow pheomelanin than in black mice. Is eumelanin solely beneficial? UVA-irradiated mice do not develop melanoma if they lack melanin (S. Premi, D.E. Brash, DNA Repair (2016), F. P. Noonan et al., Commun. 3, 884 (2012).)
- 720-726: In the paragraph 1.3.2., the authors claim: “Oxidative stress, which is defined as the imbalance between the formation of reactive oxygen species (ROS) and the cell ability to provoke an effective antioxidant action, is responsible for many biological processes including photoaging and it is involved in several important pathologies.” In chapter 3.1., the authors summarize recent reports on nano-oxidants and their role in antioxidative therapy. The authors should comment on the recent changes in the understanding of the ROS functions. “In the past two decades, ROS have undergone a shift from being molecules that invoke damage (i.e. oxidative stress) to regulating signalling pathways that impinge on normal physiological and biological responses (i.e. redox biology).” Schieber M, Chandel NS., Curr Biol. 2014; 24(10).
- 727: Eumelanin, not melanin. Pheomelanin is phototoxic.
- 873-874: “[…] PDA, plays a primary role in and treatment of cancer, selectively killing cancer cells without causing toxic effects to healthy cells”. Again, a general statement without sufficient literature research and critical approach. Is the biodistribution pattern, biodegradation process, metabolic pathway, and potential mid- and long-term implication on organs and organisms particularly the immune system of PDA and other melanin-based nanoparticles well understood? Are there any limitations, negative effects for in vivo medical applications?
- 931: Please reformulate the sentence: “[..] PTA, that when properly stimulated with light, absorb the radiation energy and convert it into heat” to e.g.: “PTA absorbs the irradiated light and convert it into heat.”
- 1513-1515: Such a general statement is very imprecise and should be avoided. Proper literature research is missing here.

Reviewer 3 Report
The authors summarize in an extensive manuscript a great number of examples of bio-applications of MNP published in the last decade, reviewing the most recent examples focusing on the multifunctionality in theranostics platforms. They summarize also the main properties of melanin and its derivative which make interesting the corresponding nanosystems for bioimaging and therapy purposes. Interesantly, the authors also highlight the applications of artificial MNP in cosmetics that is a very recent concept with very good results as observed in the bibliography afforded.
In general is a well writing review with a lot of examples, well-structured and presented. Although in some parts results a bit dense, the extensive description on the different examples justify it.
However, I would make some criticism referring to the fact that I found missing references of various experts in the biomedical application of this type of nanoparticles such as Haesin Lee and Phillip B. Messersmith, especially in the sections referring to biocoatings and surface functionalization, or others like D. Ruiz-Molina concerning synthesis of polydopamine-like nanoparticles and coatings. I suggest that the authors review the bibliography of these authors and update the one already provided.
Other issue is related to the figure 1. For me is a bit confusing… Visually is not well designed and it is difficult to try to correlate the melanin properties with its use as nanoparticles in nanomedicine and cosmetics. In the text it is well explained, but this figure should be redesigned. I would suggest make it much easier and define which properties of melanine are maintained in the MNP and their uses or applications in nanomedicine (imaging + therapy) and/or cosmetics.
I found this review very interesting a highly completed concerning the different possibilities of melanine nanoparticles in medicine and cosmetics. Apart from the mentioned suggestions made previously, I consider this review suitable for publication in Nanomaterials.
Round 2
Reviewer 2 Report
The authors correctly addressed all of the issues raised by the reviewers.